# The substrate-binding domains of the osmoregulatory ABC importer OpuA transiently interact

**Marco van den Noort[1], Panagiotis Drougkas[1,2], Cristina Paulino[1,2], Bert Poolman[1]\***

[1]Department of Biochemistry, Groningen Biomolecular Science and Biotechnology Institute, University of Groningen, Groningen, Netherlands; [2]Biochemistry Center, Heidelberg University, Heidelberg, Germany

**Abstract** Bacteria utilize various strategies to prevent internal dehydration during hypertonic stress. A common approach to countering the effects of the stress is to import compatible solutes such as glycine betaine, leading to simultaneous passive water fluxes following the osmotic gradient. OpuA from *Lactococcus lactis* is a type I ABC-importer that uses two substrate-binding domains (SBDs) to capture extracellular glycine betaine and deliver the substrate to the transmembrane domains for subsequent transport. OpuA senses osmotic stress via changes in the internal ionic strength and is furthermore regulated by the 2nd messenger cyclic-di-AMP. We now show, by means of solution-based single-molecule FRET and analysis with multi-parameter photon-by-photon hidden Markov modeling, that the SBDs transiently interact in an ionic strength-dependent manner. The smFRET data are in accordance with the apparent cooperativity in transport and supported by new cryo-EM data of OpuA. We propose that the physical interactions between SBDs and cooperativity in substrate delivery are part of the transport mechanism.

**\*For correspondence:**
b.poolman@rug.nl

**Competing interest:** The authors declare that no competing interests exist.

## eLife assessment

The OpuA Type I ABC importer uses two substrate binding domains to capture extracellular glycine betaine and present the substrate to the transmembrane domain for subsequent transport and correction of internal dehydration. This study presents **valuable** findings addressing the question of whether the two substrate binding domains of OpuA dock and physically interact in a salt-dependent manner. The single-molecule fluorescence resonance energy transfer and cryogenic electron microscopy data that are presented provide **convincing** support for the existence of a transient interaction between the substrate binding domains that depends on ionic strength, laying a foundation for future studies exploring how this interaction is involved in the overall transport mechanism.

## Introduction

All cells undergo passive fluxes of water into and out of the cell due to the semipermeable nature of biological membranes. This makes osmotic challenges a universal stress factor. Cells compensate the passive outward flux of water by accumulating compatible solutes and thereby rehydrate the cytoplasm during osmotic stress. Bacteria commonly accumulate zwitterionic compatible solutes such as glycine betaine, carnitine, TMAO and or proline (*Bremer and Krämer, 2019*; *Hoffmann and Bremer, 2017*). The import of compatible solutes induces an inflow of water that can compensate for an increase in external osmolality, thereby restoring turgor and physicochemical homeostasis of the cytoplasm (*Stadmiller et al., 2017*; *Bounedjah et al., 2012*). Compatible solute importers, sense physical properties like ionic strength or specific ion concentrations to regulate their import activity immediately

in response to osmotic stress (*van der Heide et al., 2001*; *Sikkema et al., 2020*; *Tantirimudalige et al., 2022*). The import of compatible solutes must be tightly regulated since overaccumulation can be equally lethal, causing detrimental hydrostatic pressure leading to cell lysis (*Commichau et al., 2018*; *Pham et al., 2018*; *Stülke and Krüger, 2020*). Hence, additional regulatory mechanisms can be present to control the flux of osmolytes. The second messenger cyclic-di-AMP acts as a backstop to prevent overaccumulation of compatible solutes and K⁺ (*Sikkema et al., 2020*; *Commichau et al., 2018*; *Pham et al., 2018*; *Stülke and Krüger, 2020*; *Schuster et al., 2016*; *Quintana et al., 2019*; *Corrigan et al., 2013*; *Gundlach et al., 2015*). The importance of this regulatory mechanism is highlighted by the fact that cyclic-di-AMP is both essential and toxic at elevated concentrations (*Stülke and Krüger, 2020*; *Gundlach et al., 2015*). This property makes the protein network associated with cyclic-di-AMP sensing, synthesis and breakdown valuable for finding new antimicrobial therapeutics (*Dey et al., 2017*; *Opoku-Temeng and Sintim, 2016*; *Neumann et al., 2023*).

OpuA from *Lactococcus lactis* is one of the best-studied compatible solute importers. The activity and expression of OpuA are increased by ionic strength, and are inhibited by cyclic-di-AMP (*van der Heide et al., 2001*; *Sikkema et al., 2020*; *Pham et al., 2018*; *Romeo et al., 2007*; *Mahmood et al., 2006*). OpuA belongs to the type I subfamily of ABC-importers. It is a tetrameric protein complex consisting of two different types of subunits: two cytosolic OpuAA proteins consisting of a nucleotide-binding domain (NBD) and a cyclic-di-AMP-sensing (CBS) domain, and two OpuABC proteins, consisting of a transmembrane domain (TMD), a scaffold domain and an extracellular single substrate-binding domain (SBD; *Figure 1A*). The NBDs hydrolyze ATP to fuel transport and sense ionic strength through interactions of a cationic helix-turn-helix motif with the negatively charged membrane (*Sikkema et al., 2020*). The SBD is a two-lobed protein that binds glycine betaine with micromolar affinity and delivers it to the transporting unit. In contrast to the covalently linked SBDs found in OpuA, substrate binding units in many other ABC importers are expressed as an independent soluble or lipid-anchored protein, called then a substrate-binding protein (SBP; *van der Heide and Poolman, 2002*).

There is much interest in understanding the transport mechanisms of ABC-transporters (*ter Beek et al., 2014*; *Oldham et al., 2008*). In the case of type I ABC-importers, pioneering studies have been performed on the maltose and histidine import systems, MalFGK$_2$-E and HisQMP$_2$, respectively. In both transporters, the soluble substrate-loaded SBP (MalE or HisQ) is needed to fully stimulate ATPase activity, although the SBP can also bind the TMDs and slightly stimulate ATPase activity in the absence of substrate (*Mächtel et al., 2019*; *Chen, 2013*; *Heuveling et al., 2014*; *Sippach et al., 2014*; *Ames et al., 1996*; *Bao and Duong, 2012*). The binding of MalE to the TMDs is tight in the presence of inhibiting nucleotides and goes through cycles of docking and release under transport conditions (*Bao and Duong, 2012*; *Chen et al., 2001*). There are no indications that either the MalE or HisQ proteins cooperate together. In fact, empty MalE SBPs and SBDs from the amino acid importer GlnPQ have been shown to compete with their substrate-bound equivalents for binding to the TMDs (*Merino et al., 1995*; *Gouridis et al., 2015*).

In contrast to MalE SBP and the SBDs of GlnPQ, the two SBDs in OpuA act cooperatively during transport. The cooperativity increases the import efficiency to overcome osmotic stress as quickly as possible (*Biemans-Oldehinkel and Poolman, 2003*). Despite the beauty of the cooperative mechanism, the molecular basis for the cooperativity in OpuA remains unresolved. On the contrary, MalE from *E. coli*, ProX from *Synechococcus spp.*, ArtJ from *Thermatoga maritima*, and the SBPs TakP and TM0322 from the family of tripartite ATP-independent periplasmic (TRAP) transporters can form dimers in solution, but there is no data on the functional role of the dimeric states (*Cuneo et al., 2008*; *Gonin et al., 2007*; *Richarme, 1983*; *Ford et al., 2022*; *Luchansky et al., 2010*; *Ruggiero et al., 2014*). A substrate channel through a dimer of TakP SBPs is observed in the crystal structure of the proteins (*Gonin et al., 2007*), which provides a possibility for cooperativity between two TakP SBPs.

Single-molecule FRET (smFRET) is an indispensable biophysical tool to study transient interdomain interactions (*Lerner et al., 2018*). The time resolution of solution-based smFRET is generally limited by the time that the protein resides in the confocal volume of the laser, which is usually in the millisecond range. However, the new analysis tool multi-parameter photon-by-photon hidden Markov modeling (mpH2MM) enables to study different states of the protein within a single burst (*Harris et al., 2022*; *Pirchi et al., 2016*). Furthermore, mpH2MM provides valuable insights into the transition rate constants between short-lived states.

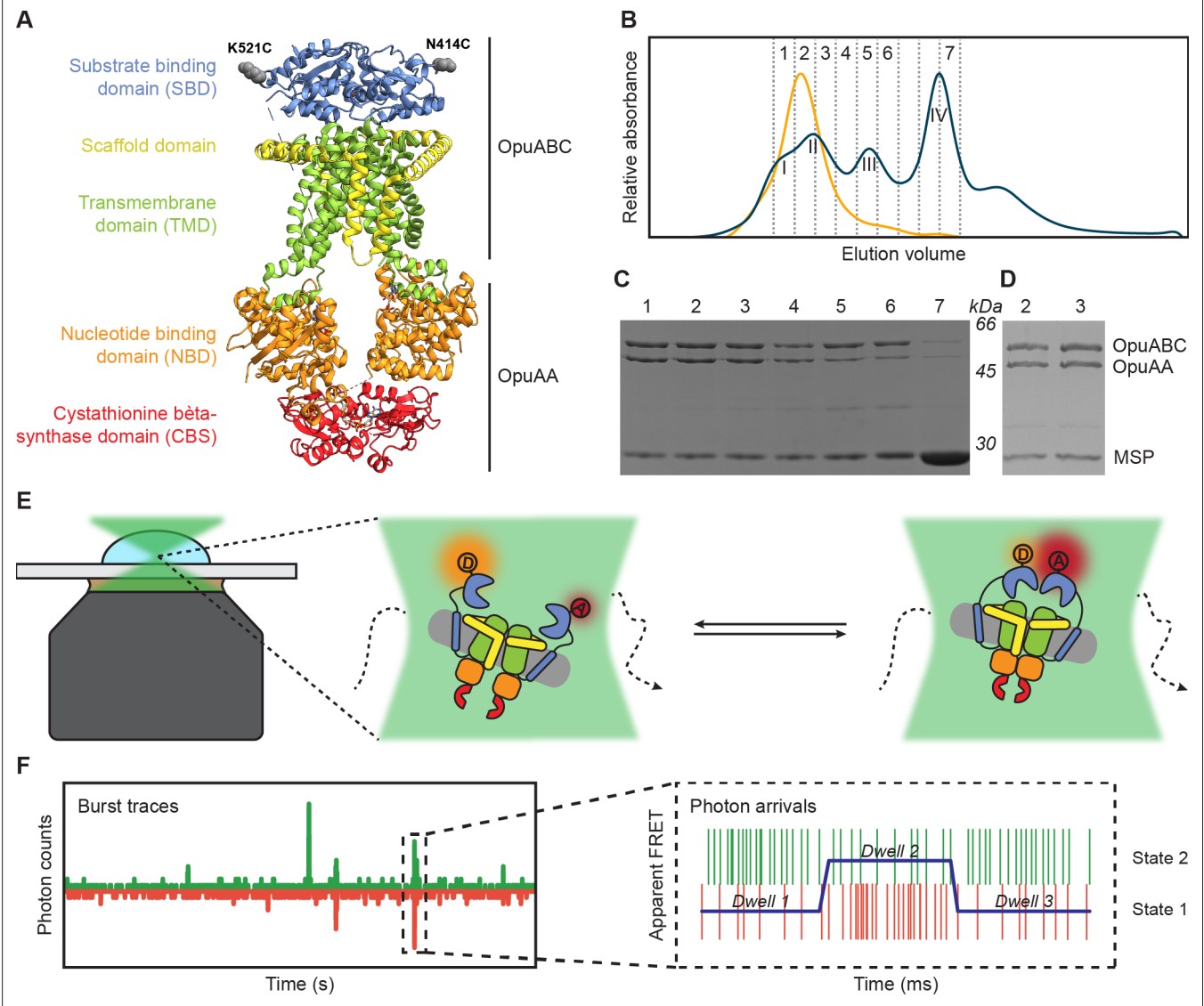

**Figure 1.** Experimental approach for measuring interdomain dynamics in the SBDs of OpuA. (**A**) A cryo-EM structure of OpuA (PDB: 7AHH). Mutations K521C and N414C are highlighted as grey spheres. (**B**) Size-exclusion chromatography profiles of OpuA-nanodiscs that were purified according to the previously described protocol (*Sikkema et al., 2020*) (blue) or according to the new protocol that is described here (yellow). The Latin numbers refer to the four different nanodisc species as is described in the first paragraph of the Results section. The other numbers refer to the elution fractions that were loaded on an SDS-PAA gel. (**C**) SDS-PAA gel with the size exclusion fractions of the blue line in (**B**). (**D**) SDS-PAA gel with the size-exclusion fractions of the yellow line in (**B**). (**E**) A schematic representation of how confocal, solution-based smFRET was used to study different states of the SBDs. (**F**) A representation of a fluorescent burst time trace, displaying the photon counts in the donor (green) and acceptor (red) detection channel over time (left). The zoom-in is a representation of a photon time trace, in which single photons are represented as lines and the most likely state path from the *Viterbi* algorithm in mpH2MM as a blue line.

The online version of this article includes the following source data and figure supplement(s) for figure 1:

**Source data 1.** Raw SDS-PAA gel images of *Figure 1C and D*.

**Source data 2.** Labelled SDS-PAA gel images of *Figure 1C and D*.

**Figure supplement 1.** Enzyme-coupled ATPase assay of wildtype OpuA (circles, grey shading), OpuA-K521C (squares, blue shading) and OpuA-N414C (triangles, yellow shading).

 We designed cysteine mutations in the SBD of OpuA to study interdomain dynamics in the full-length transporter (*Sikkema and Poolman, 2021*). The positions of the cysteines do not interfere with docking of the SBDs to the TMDs, and they do not affect the transport. We further improved the purification, nanodisc reconstitution and labeling of OpuA to obtain a sample suitable for single-molecule

studies. Next, we studied the dynamics of the SBDs in OpuA with sub-millisecond time resolution. The observed dynamics were complemented by new structural data of OpuA obtained by cryo-EM.

## Results

### Optimization of OpuA purification and nanodisc reconstitution

Previously, a protocol for the purification of OpuA and subsequent reconstitution in MSP-based nanodiscs was used to obtain cryo-EM structures of the protein complex. This protocol was suitable for obtaining high-resolution 3D-images, but the size-exclusion chromatography (SEC) profile showed that the nanodisc sample consisted of protein complexes with different molecular weights (*Figure 1B*). The resolution of the Superdex 200 Increase 10/300 column is not high enough to fully separate the different macromolecular species. Based on SDS-PAGE and electron microscopy analyses, the first peak in the SEC profile corresponds to OpuA-containing lipid structures that are distinct from OpuA nanodiscs (second peak) (*Karasawa et al., 2013*). The second peak has both subunits of OpuA (OpuAA and OpuABC) in the nanodiscs. The third peak represents OpuA nanodiscs with one or two OpuAA subunits lost. The fourth peak consists of lipid nanodiscs lacking any OpuA components (*Figure 1C*). Because the first three peaks overlap, it was not possible to obtain monodisperse OpuA nanodiscs. This is acceptable for cryo-EM studies but complicates the analysis of protein dynamics by smFRET. Therefore, we optimized the purification and reconstitution on various aspects, and we were able to achieve monodisperse OpuA nanodiscs (see Materials and methods; *Figure 1B and D*). The new procedure yielded OpuA nanodiscs with similar glycine betaine-, KCl-, and ATP-dependent activity profiles as before (*Figure 1—figure supplement 1*). Furthermore, the absolute activity was constant between different replicates and was not reduced after storage at –80 °C.

### Purification and labeling of OpuA cysteine variants

smFRET is a powerful tool for studying membrane protein dynamics (*Lerner et al., 2018*; *Bartels et al., 2021*). In order to study single-molecule dynamics of OpuA, we made use of thiol-reactive, maleimide-based donor and acceptor fluorophores Alexa Fluor 555 and Alexa Fluor 647, respectively, that were linked to engineered cysteines in natively cys-less OpuA. We record fluorescent bursts of a few milliseconds in duration, which come from single molecules that transit through the focal spot of a confocal microscope (*Figure 1E and F*). Every molecule gets excited by a pulsed laser that alternates between a donor and an acceptor excitation wavelength at a pulse rate of 40 MHz. Consequently, every molecule gets excited many times. The relative fluorescent intensity of the fluorophores after excitation of the donor is used to calculate an apparent FRET efficiency for each burst ($E^*$). The real FRET efficiency ($E$) is obtained by applying the usual correction factors (see Materials and methods). The fluorescence after excitation of the acceptor is used together with the total fluorescence after donor excitation to calculate a stoichiometry ($S$) of fluorescence. The Python-based package FRET-bursts was used to filter out molecules that were labeled with only donor ($S \approx 1$) or only acceptor ($S \approx 0$) fluorophores (*Figure 2—figure supplement 1*; *Ingargiola et al., 2016b*).

Several double-cysteine variants have been constructed in a soluble variant of the SBD (*Sikkema and Poolman, 2021*). The cysteines were designed at the back of the protein, such that the attached labels would not interfere with SBD docking onto the TMDs during the translocation process. Since full-length OpuA consists of two identical OpuABC and two OpuAA proteins, a single cysteine was introduced per SBD (OpuABC) (*Figure 1A and E*). We used the positions Lys-521 and Asn-414 to

**Table 1.** ATPase activity of OpuA variants K521C and N414C before and after labeling with maleimide dyes[*].

|  | Activity unlabeled variant (min⁻¹) | Activity labeled variant (min⁻¹) |
|---|---|---|
| K521C | 342+/-71[†] | 263+/-52 |
| N414C | 332+/-25 | 246+/-55 |

[*]Buffer conditions: 50 mM HEPES-K pH 7.0, 100 μM glycine betaine, 10 mM Mg-ATP plus 600 mM KCl.

[†]The errors refer to the standard deviation over at least two measurements with different protein purifications and membrane reconstitutions, each consisting of three technical replicates.

introduce single cysteine mutations in the full-length transporter. These variants are used to study inter-SBD dynamics in the FRET range of 3–10 nm (*Figure 1E*).

Both mutants purified very well and retained full ATPase activity in the presence of 1 mM DTT (*Figure 1—figure supplement 1*). They retained 70–80% of their original activity when labeled with the Alexa fluorophores (*Table 1*). The labeling efficiency varied between 60% and 90% across different purifications.

## SBD dynamics under high ionic strength conditions

An SBD of OpuA has limited space to diffuse because it is connected to the TMD via a transmembrane anchoring helix and a soluble linker of eleven amino acids long. Theoretically, the non-docked SBDs can sample inter-domain distances that would cover the entire FRET range from 3 to 10 nm, with low FRET values corresponding to long distances, whereas high FRET values are indicative of shorter distances. The mobility of the undocked SBDs will be higher than the diffusion of the whole complex, allowing the sampling of varying interdomain distances within a single burst. However, these dynamic variations are subsequently averaged to a singular FRET value during FRET calculations for each burst, and may appear as a single low FRET state in the FRET histograms.

Both OpuA-N414C and OpuA-K521C were first analyzed in a high salt condition without substrate or Mg-ATP (50 mM HEPES-K pH 7.0, 600 mM KCl). As expected, most FRET bursts fall within a low-FRET population when OpuA is in the apo state (*Figure 2A and B*; upper panels). However, the FRET distributions tail towards higher FRET values, especially in OpuA-K521C. We performed a burst variance analysis (BVA) to qualitatively assess if the broad FRET distribution represents different fixed states or a mixture of states that can alternate within a single burst (see Materials and methods; *Torella et al., 2011*). A large part of the FRET population exhibits within-burst dynamics, which was higher than expected from shot noise alone (*Figure 2A and B*; third panel). We then analyzed the data by multi-parameter photon-by-photon hidden Markov modeling (mpH2MM; *Harris et al., 2022*). This new approach uses raw photon data from both donor and acceptor excitation periods to find the number of hidden Markov states that best describe the data. A Viterbi algorithm identifies the optimal state path within bursts. Each state within a burst is then considered separately as a dwell (*Figure 1F*).

The FRET data of both OpuA variants is best explained by a four-state model (*Figure 2A and B*; fourth and fifth panel). Two of the four states represent donor-only (S≈1) or acceptor-only (S≈0) dwells. The full bursts belonging to donor-only and acceptor-only molecules were excluded prior to mpH2MM. This means that some molecules transit to a donor-only or acceptor-only state within the burst period, which most likely reflects blinking or bleaching of one of the fluorophores. These donor-only and acceptor-only states were also excluded during further analysis. The other two states reflect genuine FRET dwells that were analyzed by mpH2MM. They represent different conformations of the SBDs. In other words, the SBDs do not only freely diffuse in the absence of substrate and Mg-ATP (most likely represented by the low-FRET state with a high burst variance), but they also adopt a conformation where they move closer together. The transition rate constants from the high- to the low-FRET state was 193 s$^{-1}$ for OpuA-N414C and 173 s$^{-1}$ for OpuA-K521C. This is in the range of a burst size of 5–10ms and indicates that some proteins moved from one state to the other while being excited in the confocal volume. This also explains at least a part of the elevated burst variance in the BVA plots. Within-burst transitions from low- to high-FRET were less frequent than those from high- to low-FRET, making the estimated transition rate constants below 100 s$^{-1}$ less accurate.

Assuming that the low-FRET state represents a situation where the SBDs diffuse freely, we note that the mean FRET efficiency for OpuA-K521C is higher (0.18) than for OpuA-N414C (0.10). Position 414 is located farther from the transmembrane helix that anchors the SBD to the transporter. Hence, N414C has a larger sampling space and the two dyes can move farther apart, which explains the difference in mean FRET efficiency between OpuA-K521C and OpuA-N414C.

## SBD dynamics under turnover conditions

OpuA transports glycine betaine in the presence of Mg-ATP and at high ionic strength. The transport-competent docking of the SBD onto the TMDs has been resolved by cryo-EM in a pre-hydrolysis state by using Mg-ATP and glycine betaine in combination with the E190Q mutation, which prevents OpuA from hydrolyzing ATP (*Sikkema et al., 2020*). It is unknown whether SBD docking also occurs in other states of the transport cycle.

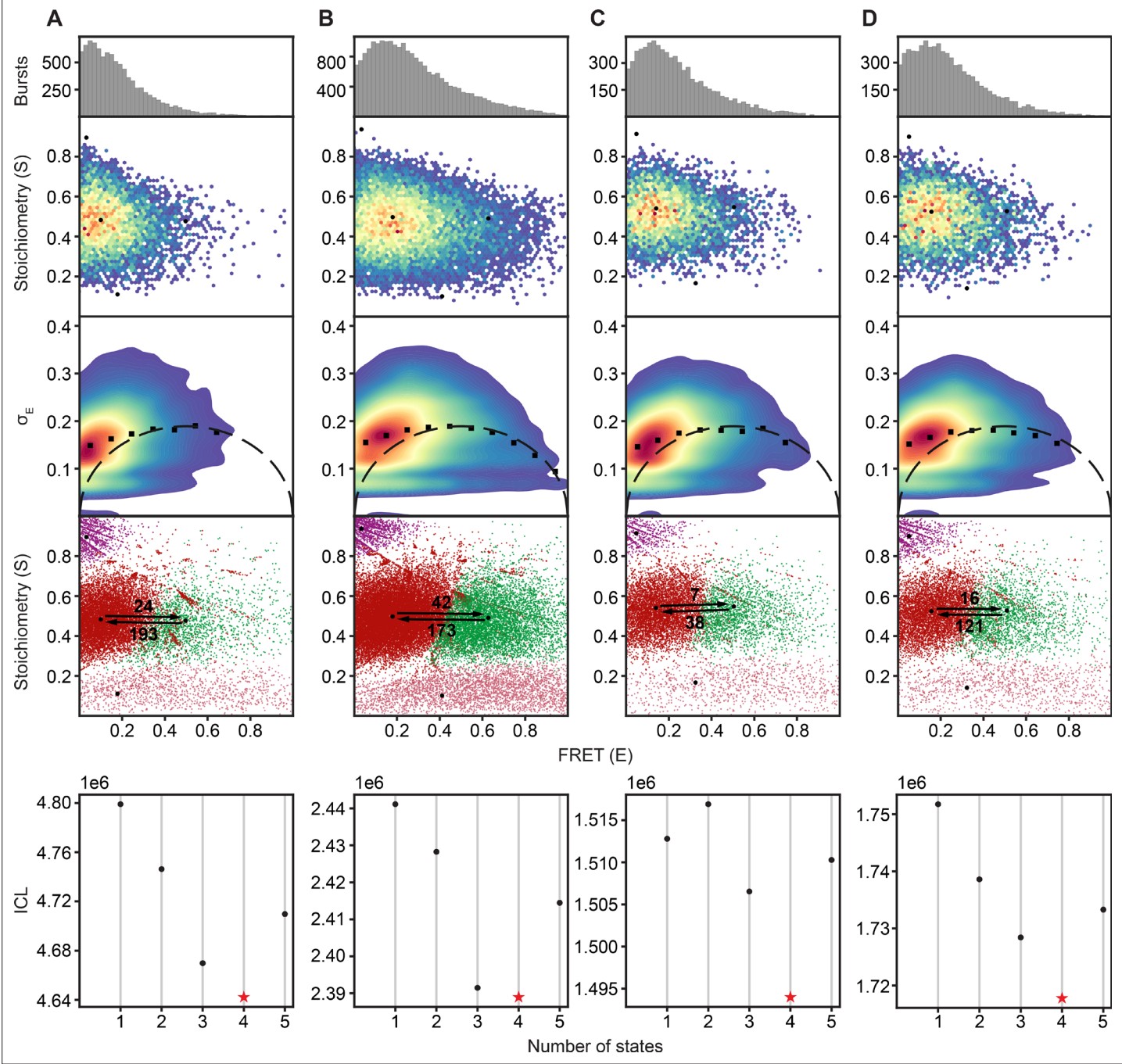

**Figure 2.** The SBDs of OpuA sample two dynamic FRET states. The proteins were analyzed in 50 mM HEPES-K pH 7.0, 600 mM KCl with the following additions: (**A**) OpuA-N414C without further additions. (**B**) OpuA-K521C without further additions. (**C**) OpuA-E190Q-K521C with 20 mM Mg-ATP plus 100 μM glycine betaine. (**D**) OpuA-K521C with 20 mM Mg-ATP. From top to bottom: (i) FRET histogram showing the corrected bursts that were selected after removing donor-only and acceptor-only bursts. (ii) 2D E-S histogram showing the same data as in (i). Black dots represent the average value of each state after mpH2MM and after application of the correction factors. (iii) Burst variance analysis of the same burst data as in (i). The standard deviation of FRET in each burst is plotted against its mean FRET. Black squares represent average values per FRET bin. The black dotted line shows the expected standard deviation in the absence of within-burst dynamics. (iv) E-S scatter plot of the corrected dwells. Dwells are colored on the basis of the assigned state of the chosen mpH2MM model. Black dots represent the average value of each state and the numbers at the arrows show transition rate constants (s⁻¹) between the two FRET states. (v) Plot of the ICL-values for each final model. The model used in the analysis is shown as a red star.

The online version of this article includes the following figure supplement(s) for figure 2:

**Figure supplement 1.** Typical 2D E-S histograms of FRET bursts after γ-correction and correction for leakage and crosstalk.

**Table 2.** The FRET (E) and Stoichiometry (S) of the FRET states of OpuA-K521C, their relative abundance and the transition rate constants between the states.

| Condition* | Low-FRET state (L) | | | High-FRET state (H) | | | Transition rate (s⁻¹) | | Number of FRET dwells |
|---|---|---|---|---|---|---|---|---|---|
| | S | E | % dwells | S | E | % dwells | L-H | H-L | |
| 50 mM BIS-TRIS pH 7.0, 0 mM KCl | 0.537 | 0.250 | 64.1 | 0.521 | 0.749 | 35.9 | 118 | 153 | 16,788 |
| 0 mM KCl | 0.531 | 0.248 | 66.9 | 0.519 | 0.742 | 33.1 | 123 | 175 | 27,047 |
| 0 mM KCl, 100 µM glycine betaine | 0.532 | 0.245 | 68.3 | 0.517 | 0.781 | 31.7 | 120 | 166 | 17,364 |
| 50 mM KCl | 0.526 | 0.225 | 68.0 | 0.516 | 0.735 | 32.0 | 88 | 153 | 14,708 |
| 100 mM KCl | 0.525 | 0.202 | 74.6 | 0.513 | 0.691 | 25.4 | 78 | 178 | 13,532 |
| 200 mM KCl | 0.519 | 0.183 | 81.4 | 0.518 | 0.663 | 18.6 | 41 | 153 | 16,454 |
| 400 mM KCl | 0.516 | 0.168 | 82.0 | 0.518 | 0.620 | 18.0 | 28 | 120 | 15,946 |
| 600 mM KCl | 0.497 | 0.180 | 82.6 | 0.490 | 0.628 | 17.4 | 42 | 173 | 38,604 |
| 600 mM KCl (V149Q) | 0.495 | 0.164 | 81.9 | 0.492 | 0.608 | 18.1 | 31 | 112 | 11,034 |
| 1000 mM KCl | 0.505 | 0.148 | 79.9 | 0.511 | 0.591 | 20.1 | 26 | 109 | 12,666 |
| 600 mM KCl, 20 mM Mg-ATP | 0.524 | 0.157 | 81.5 | 0.527 | 0.510 | 18.5 | 16 | 121 | 12,061 |
| 600 mM KCl, 20 mM Mg-ATP, 100 µM glycine betaine | 0.515 | 0.155 | 78.5 | 0.511 | 0.507 | 21.5 | 0 | 52 | 17,155 |
| 600 mM KCl, 20 mM Mg-ATP, 100 µM glycine betaine (E190Q) | 0.540 | 0.141 | 81.4 | 0.547 | 0.505 | 18.6 | 7 | 38 | 10,832 |
| 600 mM KCl, 20 mM Mg-ATP, 100 µM glycine betaine, 500 µM *ortho*vanadate | 0.517 | 0.154 | 78.9 | 0.513 | 0.529 | 21.5 | 17 | 87 | 21,052 |
| 600 mM KCl, 50 mM glutamate, 50 mM arginine | 0.510 | 0.162 | 79.9 | 0.513 | 0.549 | 20.1 | 19 | 133 | 12,722 |

*All conditions, except the first, contained 50 mM HEPES-K pH 7.0.

The online version of this article includes the following source data for table 2:

**Source data 1.** Burst variance analysis and mpH2MM of the FRET bursts of OpuA-K521C (variants) in different buffers.

The SBD dynamics were tested in the presence of Mg-ATP, under turnover conditions (20 mM Mg-ATP, 100 µM glycine betaine), in a pre-hydrolysis inhibited state (20 mM Mg-ATP, 100 µM glycine betaine, OpuA-E190Q) and in a post-hydrolysis inhibited state (20 mM Mg-ATP, 100 µM glycine betaine, 500 µM *ortho*-vanadate). The mean FRET efficiency of the high-FRET states in OpuA-K521C shifted by 0.1 FRET unit in all conditions (*Figure 2C and D*; *Table 2*). For OpuA-N414C, the high-FRET state got less populated but did not shift (*Table 3*). It was hard to estimate the mean FRET efficiency of this state because of the low number of dwells. Nonetheless, it is clear that the changes in the FRET state are not likely to reflect a change in docking, since the same effect is observed when Mg-ATP is present without glycine betaine (*Figure 2D*).

## SBD dynamics in a mutant with reduced docking efficiency

To further corroborate the notion that neither the low- nor high-FRET states reflect a docked state, we designed OpuA-V149Q in the OpuA-K521C background to alter the docking interface between the SBD and TMDs (*Figure 3A*). Each valine-149 located on the TMD of the two OpuABC subunits

**Table 3.** The FRET (E) and Stoichiometry (S) of the FRET states of OpuA-N414C, their relative abundance, and the transition rate constants between the states.

| Condition* | Low-FRET state (L) | | | High-FRET state (H) | | | Transition rate (s⁻¹) | | Number of FRET dwells |
|---|---|---|---|---|---|---|---|---|---|
| | S | E | % dwells | S | E | % dwells | L-H | H-L | |
| 0 mM KCl | 0.527 | 0.099 | 72.2 | 0.543 | 0.509 | 27.8 | 15 | 68 | 12,162 |
| 600 mM KCl | 0.483 | 0.100 | 88.5 | 0.476 | 0.497 | 11.5 | 24 | 193 | 18,130 |
| 600 mM KCl, 20 mM Mg-ATP | 0.548 | 0.090 | 94.4 | 0.536 | 0.456 | 5.6 | 2 | 141 | 10,447 |
| 600 mM KCl, 20 mM Mg-ATP, 100 µM glycine betaine | 0.538 | 0.087 | 95.7 | 0.555 | 0.499 | 4.7 | 2 | 59 | 14,898 |
| 600 mM KCl, 20 mM Mg-ATP, 100 µM glycine betaine (E190Q) | 0.479 | 0.106 | 94.8 | 0.464 | 0.540 | 5.2 | 25 | 317 | 8024 |
| 600 mM KCl, 20 mM Mg-ATP, 100 µM glycine betaine, 500 µM orthovanadate | 0.538 | 0.085 | 94.6 | 0.578 | 0.568 | 5.4 | 5 | 186 | 7236 |

*All conditions contained 50 mM HEPES-K pH 7.0.

The online version of this article includes the following source data for table 3:

**Source data 1.** Burst variance analysis and mpH2MM of the FRET bursts of OpuA-N414C (variants) in different buffers.

interacts with a different lobe of a single docked SBD. V149Q was designed as a mild mutation that would reduce docking efficiency and thereby substrate loading, but leave the ionic strength sensing in the NBD and the binding of glycine betaine and ATP intact. Accordingly, a reduced docking efficiency should result in a lower absolute glycine betaine-dependent ATPase activity. At the same time the responsiveness of the system to varying KCl, glycine betaine, or Mg-ATP concentrations should not change. OpuA-V149Q-K521C exhibited a two- to three-fold reduction in glycine betaine-dependent ATPase activity, whereas the futile hydrolysis of ATP in the absence of substrate was unaffected (*Figure 3B*). The glycine betaine-dependent ATPase activities of wild type OpuA and OpuA-V149Q-K521C are comparable when they are plotted relative to their maximal activities (*Figure 3C*). The same holds for the dependencies on Mg-ATP and ionic strength (KCl concentration) (*Figure 3—figure supplement 1*). This confirms that the V149Q mutation affects the docking interface but not any other property of OpuA.

We then studied OpuA-V149Q-K521C by smFRET at apo conditions, similarly as described above, in a high salt buffer (50 mM HEPES-K pH 7.0, 600 mM KCl) without glycine betaine or Mg-ATP. The FRET dwells of OpuA-V149Q-K521C also belong to two states with very similar transition rate constants and relative abundances compared to OpuA-K521C (*Figure 3D*; *Table 2*). Therefore, it is highly unlikely that any of the FRET states represents a conformation in which one SBD is docked.

## KCl and glycine betaine-dependent changes in FRET states

Because the change in FRET in the presence of Mg-ATP does not reflect a change in docking dynamics, we tested whether the difference in ionic strength between a buffer with or without Mg-ATP was causing the shift. FRET bursts of OpuA-K521C were recorded in the presence of KCl concentrations, ranging from 0 to 1000 mM. Notably, the mean FRET efficiency of the high-FRET state gradually shifted from 0.74 to 0.59 in the salt range of 0–1000 mM KCl (*Figure 4A*; *Table 2*). This is indicative of a structural change. Also the percentage of dwells that reside in the high-FRET state changed from 33% to 17% (*Table 2*). Interestingly, the transition rate constant from the high- to low-FRET population did not differ much between high and low KCl, whereas the transition rate constant from the low- to high-FRET population decreased from around 120 s⁻¹ to 26 s⁻¹ (*Table 2*). In other words, the increase in high-FRET dwells is caused by an increased tendency to go from low to high-FRET, while the tendency to go from high to low-FRET does not change.

The high-FRET state of OpuA-N414C decreased from 28% to 12% when the KCl concentration was increased from 0 to 600 mM (*Figure 4B*; *Table 3*). However, this was not accompanied by a change in FRET or an increase in the transition rate constant from low- to high-FRET, but rather a decrease in the transition rate constant from high- to low-FRET (*Table 3*). Assuming that the high-FRET states in OpuA-K521C and OpuA-N414C are a result of the same conformation, this could mean that there

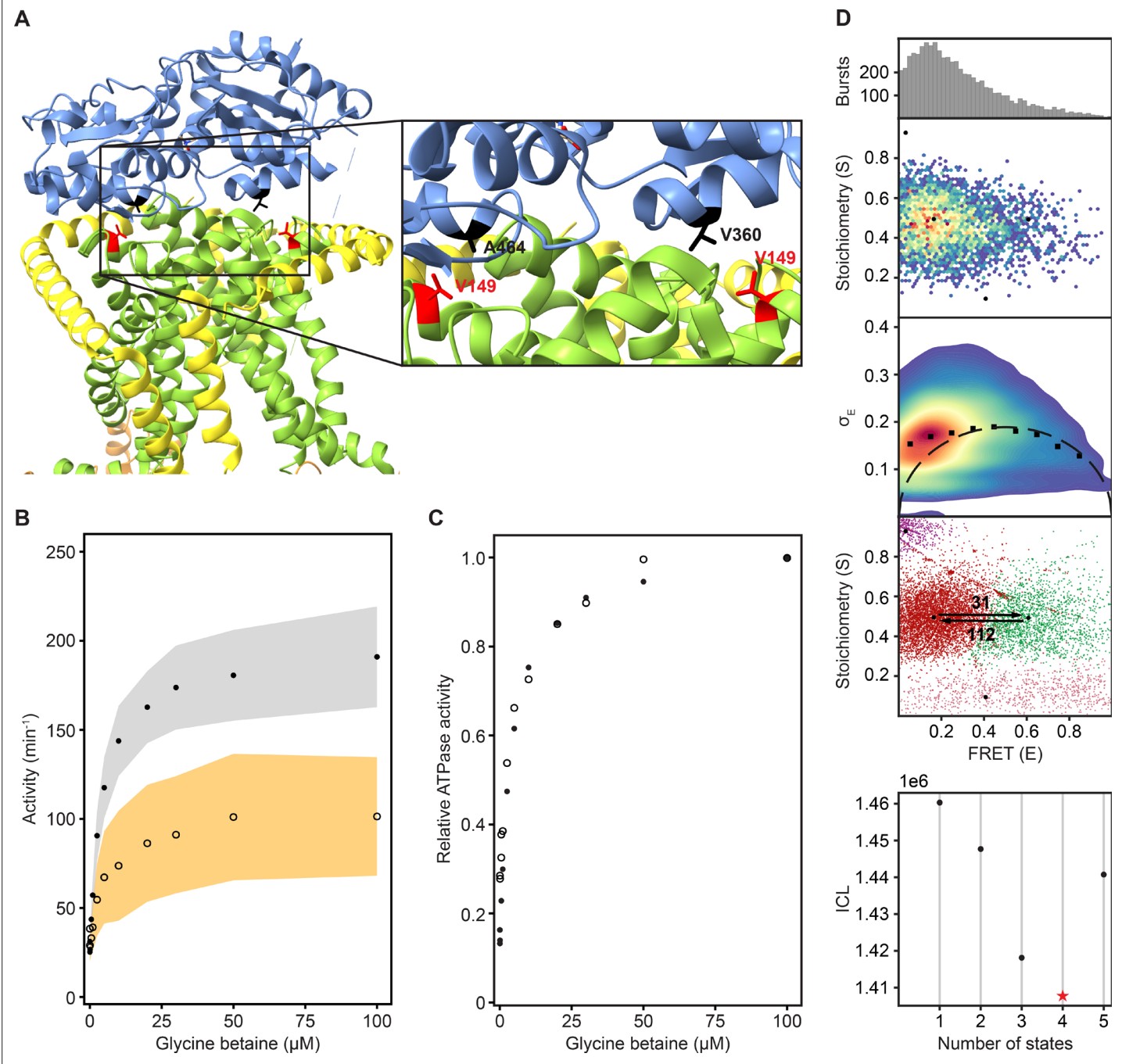

**Figure 3.** Reduction in SBD docking efficiency does not affect the two dynamic FRET states. (**A**) A cryo-EM structure of OpuA (PDB: 7AHH) highlighting Val-149 and the two most important residues in the vicinity of Val-149. Coloring of the domains is similar to that in *Figure 1A*. (**B**) Results of an enzyme-coupled ATPase assay for OpuA-WT (filled circles, grey) and OpuA-V149Q-K521C (open circles, yellow). Each sample contains 50 mM HEPES-K pH 7.0, 450 mM KCl, 10 mM Mg-ATP, 4 mM phospho*enol*pyruvate, 600 μM NADH, 2.1–3.5 U of pyruvate kinase, and 3.2–4.9 U of lactate dehydrogenase. Standard deviation over at least two measurements with different protein purifications and membrane reconstitutions, each consisting of three technical replicates is represented as shaded areas. (**C**) Results of (**B**) represented as activity relative to the activity at 100 μM glycine betaine. (**D**) smFRET results for OpuA-V149Q-K521C in 50 mM HEPES-K pH 7.0, 600 mM KCl. From top to bottom: (i) FRET histogram showing the corrected bursts that were selected after removing donor-only and acceptor-only bursts. (ii) 2D E-S histogram showing the same data as in (i). Black dots depict the average value of each state after mpH2MM and after application of the correction factors. (iii) Burst variance analysis of the same burst data as in (i). The standard deviation of FRET in each burst is plotted against its mean FRET. Black squares represent average values per FRET bin. Black dotted line shows the expected standard deviation in the absence of within-burst dynamics. (iv) E-S scatter plot of the corrected dwells. Dwells are colored on the basis of the

*Figure 3 continued on next page*

*Figure 3 continued*

assigned state of the chosen mpH2MM model. Black dots represent the average value of each state and the numbers at the arrows show transition rate constants (s⁻¹) between the two FRET states. (v) Plot of the ICL-values for each final model. The model used in the analysis is shown as a red star.

The online version of this article includes the following figure supplement(s) for figure 3:

**Figure supplement 1.** Enzyme-coupled ATPase assay for wildtype OpuA (squares, grey shading) and OpuA-V149Q-K521C (circles, yellow shading).

is more conformational flexibility across different conditions between the two Cys-521 positions than between the Cys-414 positions.

Notably, in contrast to many other ABC transporters, the SBDs are covalently linked to the TMD. Thus, the movement of the SBDs in OpuA is constrained by a short linker consisting of only 11 amino acids and a hydrophobic transmembrane anchoring sequence. These sequences are not conserved among species within the same order of Lactobacillales, and the linker sequence is not predicted to form any secondary structure (*Figure 4—figure supplement 1*; *Drozdetskiy et al., 2015*). The backbone of a polypeptide in its extended conformation spans ~0.35 nm per amino acid and the radius of the SBD is around 2 nm (*Schuurman-Wolters et al., 2018*). Assuming that the position of the linker at the membrane is fixed, each SBD has a mere sampling space of half a sphere with a radius of about 5.9 nm. Hence, the apparent concentration of an SBD in this volume is 4 mM. With the sampling space overlapping, the two SBDs experience each other at millimolar concentrations (*Figure 4C*). Weak intermolecular interactions are sufficient for binding at millimolar concentration. The high-FRET state may thus reflect a dimeric state of the SBDs. A known procedure to overcome weak interactions at high protein concentrations is the addition of charged amino acids (*Schneider et al., 2011*; *Golovanov et al., 2004*). Therefore, we tested the dynamic behavior of OpuA-K521C in 50 mM HEPES pH 7.0, 600 mM KCl supplemented with 50 mM arginine plus 50 mM glutamate. The amino acids changed the high-FRET state to a mean FRET efficiency of 0.55 but did not prevent the presence of this state (*Figure 4A*; *Table 2*). Also, the high-FRET state is observed not only in HEPES but also in BIS-TRIS buffer (*Table 2*).

smFRET was also performed in the absence of KCl and with a saturating concentration of glycine betaine (100 µM). The mean FRET efficiency of the high-FRET state of OpuA-K521C increased to 0.78, which corresponds to an inter-dye distance of about 4 nm. This indicates that the dyes at the two SBDs move very close towards each other (*Figure 4A*; *Table 2*). It also shows that the high-FRET state can be sensitive to the conformation of the SBD. In high-salt conditions plus Mg-ATP, glycine betaine does not induce a significant FRET shift, indicating that the high-FRET state under high-salt conditions is not affected by SBD closing. Taken together, these data show that the high-FRET state is present in both the open-unliganded and closed-liganded states of the SBDs.

## Direct interactions between SBDs observed by single particle cryo-EM

Recent advances in both hardware and software for single-particle cryo-EM have drastically improved the resolution of protein structures, and they also provided the capability to sample the dynamic conformational landscape of proteins at a given condition. More specifically, during image processing, single particles of various orientations are aligned and their signal is combined, resulting in a 3D density map that represents their average state. Depending on the degree of flexibility, the signal of individual domains can align (static) or be averaged out (inherently flexible). The latter is what we anticipated for the SBDs of OpuA, but such low-resolution data can still provide invaluable information and serve to complement smFRET data. While they might not allow to unambiguously model the atomic structure of a protein domain, they reveal its predominant relative localization relative to the rest of the protein complex.

To gain insight on the interaction between the SBDs, cryo-EM datasets of full-length wild type OpuA were collected in two low ionic strength conditions (20 mM HEPES-K pH 7.0, supplemented with 50 mM or 100 mM KCl). After several rounds of classification (2D, 3D) and refinement, EM maps aligned at 6.2 and 7 Å, with particle stacks of 166,066 and 80,348, respectively (*Figure 4—figure supplements 2 and 3*). Due to the inherent flexibility of the SBDs, with respect to both the MSP protein of the nanodisc and the TMDs of OpuA, their resolution is limited. Furthermore, the cryo-EM reconstructions average all the particles in the final dataset, including those with a low and high FRET state. Nevertheless, in both conditions, the densities that correspond to the SBDs can be observed

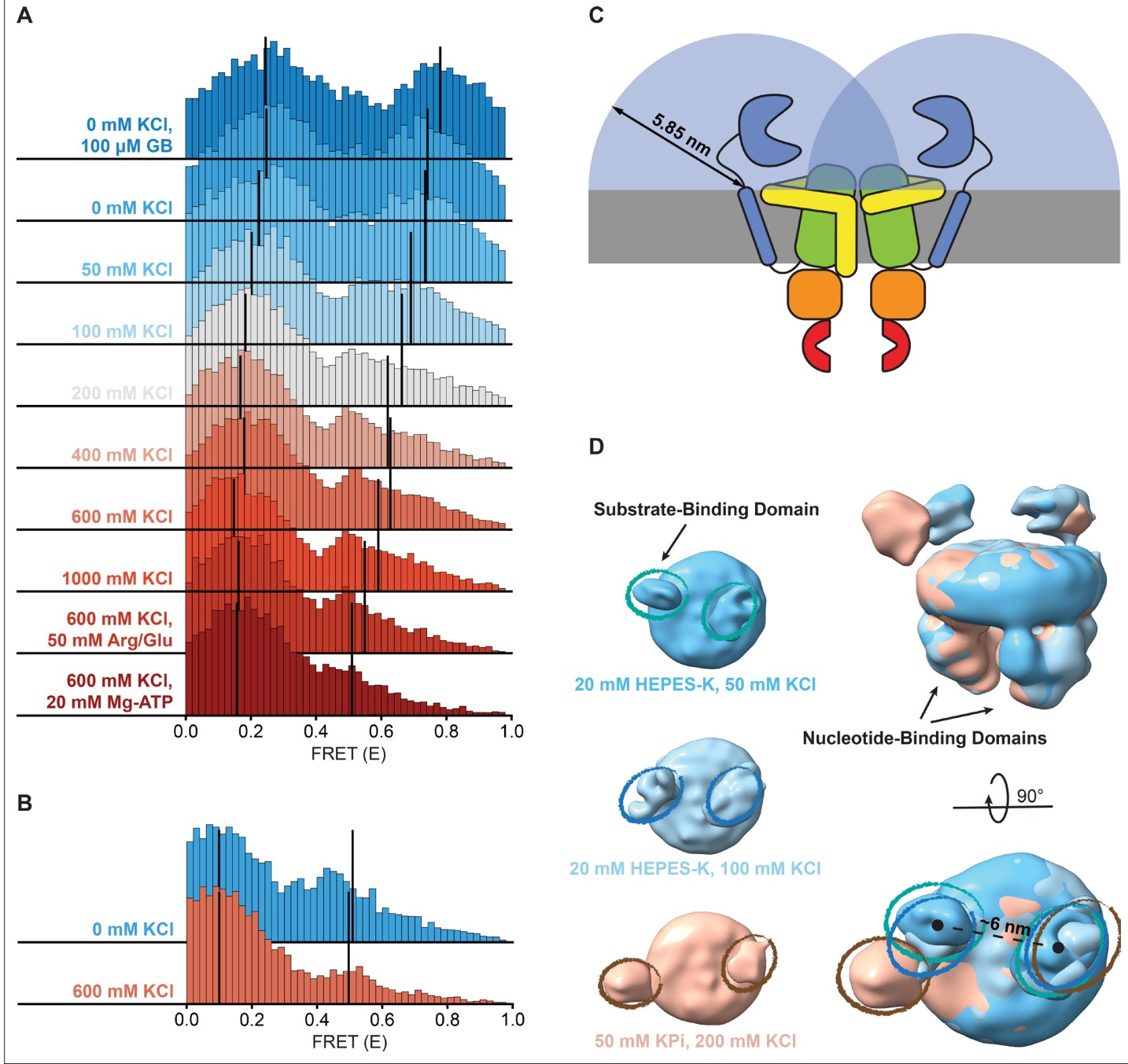

**Figure 4.** Shifts in FRET states under different conditions. (**A, B**) FRET (E) histograms of OpuA-K521C (**A**) and OpuA-N414C (**B**) corresponding to the corrected dwells of the two FRET states after mpH2MM. Black lines show the mean E of the two states. (**C**) A schematic representation of the diffusion freedom in 2D of the SBDs, shown as blue semicircles. The radius is defined as the sum of the radius of an SBD and the length of the linker region in a fully extended conformation. The sequence of the linker region was defined based on the occluded OpuA structure (PDB: 7AHD) and spans the N- and C-terminal end of the SBD and anchoring helix, respectively. (**D**) Lowpass-filtered cryo-EM density maps of OpuA-WT in three different ionic strength conditions (50 mM, 100 mM and 200 mM KCl). The top-views of each condition are horizontally aligned (left) and super-positioned (right) for better comparison. The densities corresponding to the SBDs are highlighted in teal, blue and brown ellipses, respectively.

The online version of this article includes the following figure supplement(s) for figure 4:

**Figure supplement 1.** Multiple sequence alignment of close homologs of OpuA from *L.*

**Figure supplement 2.** Image processing of OpuA wild-type in MSP1E3D1 nanodiscs in 20 mM HEPES-K pH 7.0, 50 mM KCl.

**Figure supplement 3.** Image processing of OpuA wild-type in MSP1E3D1 nanodiscs in 20 mM HEPES-K pH 7.0, 100 mM KCl.

in close proximity (*Figure 4D*). The distance between the density centers is 6 nm and align with the dimensions of an SBD, providing further evidence for physical interactions between the SBDs.

In contrast, the density derived from single particle analysis of wild type OpuA in higher ionic strength (50 mM KPi pH 7.0, supplemented with 200 mM KCl), as described by *Sikkema et al., 2020*, reveals an overall more distant localization of the SBDs with respect to each other (*Figure 4D*, left). When super-positioned, the three maps reveal a significant difference in the localization of the SBDs (*Figure 4D*, right). Given the flexibility and limited resolution, the nature of their interaction cannot be structurally characterized, but the cryo-EM data corroborate with the smFRET data.

## Discussion

The study of transient interdomain interactions in membrane-embedded proteins is challenging. Membrane proteins are unstable outside their native lipid environment and transient interactions are very hard to capture by cryo-EM or other biophysical techniques. We used smFRET to investigate the dynamics between the two SBDs of OpuA, which are covalently linked to the two TMDs in the transporter complex. The smFRET-compatible OpuA-K521C and OpuA-N414C are fully functional in ATP-, KCl-, and glycine betaine-dependent ATP hydrolysis when reconstituted into lipid nanodiscs. The majority of the recorded FRET bursts are present in a low-FRET state but exhibit within-burst dynamics according to a burst variance analysis (*Figure 2*; third panel). The bursts in the low-FRET state suggests that the protein has more than one FRET state, which are averaged out due to the limited temporal resolution of solution-based confocal microscopy. Further analysis of the data by a hidden Markov modeling approach (mpH2MM) indeed revealed a second state with a higher mean FRET efficiency and a lifetime of less than 10ms in most conditions. This is in the same range as the length of a single burst. Thus, the SBDs of OpuA not only reside in a freely diffusing low-FRET state but also in a high-FRET state where the SBDs are close to each other.

In many type I ABC importers, the SBD is expressed as a separate polypeptide, called SBP. SBPs diffuse in 3D in the bacterial periplasm or in 2D along the membrane surface, that is when they are membrane-anchored via a lipid moiety. It has been shown for several SBPs that they can dock onto the TMDs in the open-unliganded and closed-liganded states (*Ames et al., 1996*; *Bao and Duong, 2012*; *Chen et al., 2001*). Likewise, the high-FRET state could represent a state in which one of the two SBDs is docked and the other SBD is located close by. Therefore, OpuA-V149Q was designed to reduce the docking efficiency of the SBDs in OpuA (*Figure 3*). Because the relative distributions of the two FRET states are the same in OpuA-V149Q and wild type OpuA, we conclude that the high-FRET state does not represent a docked state (*Figure 3D*; *Table 2*).

Using the ATP-hydrolysis defective OpuA-E190Q, we have previously shown by cryo-EM that glycine betaine plus Mg-ATP stabilize a docked state of one of the SBDs (*Sikkema et al., 2020*). In smFRET measurements, glycine betaine plus Mg-ATP induced a shift in the mean FRET efficiency of the high-FRET states of both OpuA-K521C and OpuA-E190Q-K521C (*Figure 2C*; *Table 2*). This could indicate that a docked state is induced, which has replaced the previous high-FRET state. However, for OpuA-N414C we do not observe a shift but a reduction in the amount of dwells in the high-FRET state. Moreover, the addition of Mg-ATP alone induces similar shifts or changes in the relative amounts of the states. Therefore, it is not likely that the shift in the high-FRET state of OpuA-K521C represents a docked state of an SBD. We conclude that the spatial resolution of smFRET is not sufficient to study docking by looking at SBD-SBD distance dynamics.

We have shown that the SBDs of OpuA come close together in a short-lived state, which is responsive to the addition of glycine betaine (*Figure 4A*). Although the occurrence of the state varies between different conditions, it was not possible to negate the high-FRET state completely, not even under very high or low KCl concentrations, or in the presence of 50 mM arginine plus 50 mM glutamate (*Figure 4A and B*). To evaluate possible scenarios of interdomain interactions we consider the following: (1) The SBDs of OpuA are connected to the TMDs with very short linkers of approximately 4 nm, which limit their movement and allow the receptor to sample a relatively small volume near its docking site. (2) in low ionic strength condition OpuA-K521C displays a high FRET state with mean FRET values of 0.7–0.8, which corresponds to inter-dye distances of approximately 4 nm. (3) The high FRET state is responsive to glycine betaine, which points toward direct communication between the two SBDs. (4) The distance between the density centers of the SBDs in the cryo-EM reconstructions (based on particles with a low and high FRET state) is 6 nm, which aligns with the dimensions of an SBD

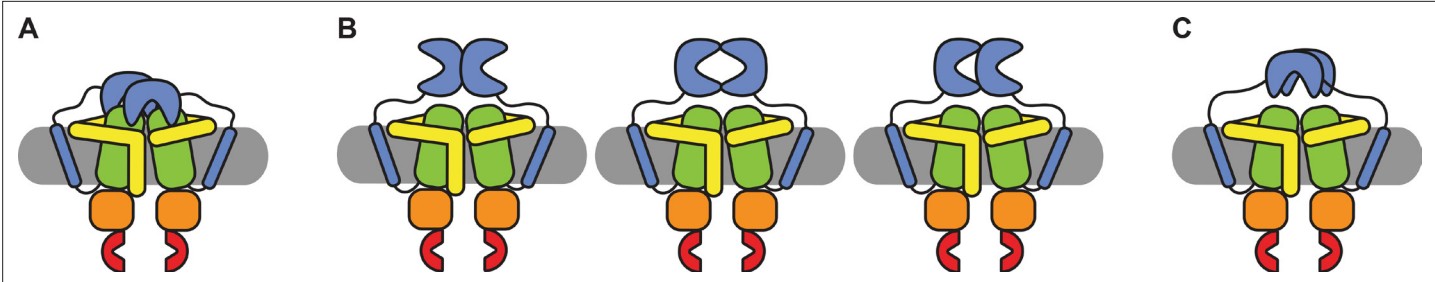

**Figure 5.** Schematic of possible states of the SBDs of OpuA. (**A**) Both SBDs dock in a non-productive manner onto the TMDs. (**B**) The SBDs interact with each other in an upright orientation either back-to-back, front-to-front or front-to-back. (**C**) The SBDs interact sideways with each other.

(length: ~6 nm, maximal width: ~4 nm). These findings collectively indicate that two SBDs interact but not necessarily in a singular conformation but possibly as an ensemble of weakly interacting states. Hence, we discuss three possible SBD-SBD interaction models to explain the high-FRET state:

Firstly, the SBDs do not physically interact but bind simultaneously to the TMDs (*Figure 5A*). Note that OpuA forms a homodimeric complex and one SBD could potentially bind to one half of the membrane-embedded transporter unit, while the other binds to the other half. This type of docking would be very different from the transport-competent state that we previously observed by cryo-EM. We find this possibility unlikely because transport-competent docking would reduce the presence of the high-FRET state, which is not what we find. Moreover, we do not observe any indication for such a state by cryo-EM in 50 mM HEPES-K pH 7.0, 50 mM KCl, whereas the high-FRET state is prominent in this condition. More specifically, a transport-competent docking would drastically reduce the flexibility of the SBDs, increasing the local resolution in the cryo-EM maps.

Secondly, the SBDs interact with each other in an upright orientation, either back-to-back, front-to-front, back-to-front or something in between (*Figure 5*). For back-to-back and back-to-front conformations, the distance between the cysteines would be less than 4 nm in both OpuA-K521C and OpuA-N414C. This cannot explain the mean FRET efficiency of the high-FRET state in both OpuA variants, which would correspond to 4.1 nm for OpuA-K521C and 4.9 nm for OpuA-N414C in the conditions with the highest mean FRET (assuming a Förster distance ($R_0$) of 0.51) (*Tables 2 and 3*). Moreover, both situations would reduce the rotational freedom of the fluorophores on the back of the SBD and hence their steady-state anisotropy, especially in the absence of KCl when the high-FRET state is more populated. However, the steady-state anisotropy is not different for conditions with 0 mM and 300 mM KCl (*Table 4*). A front-to-front conformation could reflect the observed mean FRET efficiency and at the same time would not interfere with the rotational freedom of the fluorophores. However, this conformation would not be able to dock. Since the high-FRET state is short-lived, docking should reduce the prevalence of this state, which is not what we observe (*Tables 2 and 3*).

Thirdly, the SBDs interact sideways (*Figure 5C*). In this scenario, the mean FRET efficiency of the high-FRET state can reflect the distance between the fluorophores, the fluorophores would not be

**Table 4.** Steady-state anisotropy values of free dyes and dyes which were attached to different proteins.

| | Condition* | Alexa Fluor 555[†] | Alexa Fluor 647[†] |
|---|---|---|---|
| Free dye | 50 mM KCl | 0.21 | 0.16 |
| Free dye | 300 mM KCl | 0.21 | 0.16 |
| MalE-T36C-S352C | 300 mM KCl | 0.26 | 0.22 |
| OpuA-K521C | 50 mM KCl | 0.30 | 0.28 |
| OpuA-K521C | 300 mM KCl | 0.29 | 0.27 |
| OpuA-N414C | 50 mM KCl | 0.28 | 0.24 |
| OpuA-N414C | 300 mM KCl | 0.28 | 0.23 |

*All conditions contained 50 mM HEPES-K pH 7.0.
[†]Standard deviation between triplicates was less than 0.001.

restricted in their rotational freedom, and the SBDs would be able to dock onto the TMDs. Furthermore, this would be in line with our cryo-EM data which hint to two flexible SBDs that on average move closer toward each other in low ionic strength conditions. Therefore, we consider this state type to be the most plausible explanation. It is possible that the apparent high concentration of the SBDs induces transient and not very strong interactions, explaining why the FRET can shift between different experimental conditions.

In other words, the high FRET state may comprise an ensemble of weakly interacting states rather than a singular stable conformation, resembling the quinary structure of proteins. The quinary structure of proteins is typically revealed in highly crowded cellular environments and describes the weak interactions between protein surfaces that contribute to their stability, function, and spatial organization (*Guin and Gruebele, 2019*). Despite the current study being conducted under dilute conditions, the local concentration of SBDs (~4 mM) mimics a densely populated environment and reveal quinary structure.

SBDs or SBPs that use the same transporter for docking and substrate delivery compete for the same docking site. When different SBDs with varying substrate scopes compete, or when substrate-free docking is prevalent, transport rates can be significantly reduced (*Merino et al., 1995*; *Gouridis et al., 2015*). Interestingly, the SBDs of OpuA cooperate to enhance transport activity (*Biemans-Oldehinkel and Poolman, 2003*). The interdomain interactions between SBDs reduce their diffusional freedom and keep them very close to the TMDs. Therefore, we think that the data presented in this paper elegantly explain how the SBDs of OpuA cooperate. The weak interactions between SBDs keep the SBDs closer to the TMDs, which allows for more efficient delivery and transport of glycine betaine. As soon as one SBD has delivered its substrate and is released from the TMDs, the other SBD is in very close proximity to dock onto the TMDs for a second round of transport. The interactions also raise the apparent concentration of the SBDs near the TMDs, increasing the binding propensity of an SBD with the TMDs. This also explains why the $V_{max}$ of OpuA with one inactivated and one functional SBD is not different to wild type OpuA, whereas one missing SBD reduces the $V_{max}$ twofold (*Biemans-Oldehinkel and Poolman, 2003*).

In conclusion, the SBDs of OpuA transiently interact in a docking competent conformation, explaining the cooperativity between the SBDs during transport. The conformation of this interaction is not fixed but differs substantially between different conditions. Dimer formation of SBPs has been described for a variety of proteins from different structural clusters of SBPs (*Cuneo et al., 2008*; *Gonin et al., 2007*; *Richarme, 1983*; *Ford et al., 2022*; *Luchansky et al., 2010*; *Ruggiero et al., 2014*; *Zheng et al., 2011*; *Shi et al., 2009*; *Sippel et al., 2011*). It could well be that cooperativity and transient interactions between SBDs is more common than previously anticipated, not only in covalently linked SBDs but also soluble SBPs. It is important to note that the concentration of SBPs in the periplasm of *E. coli* can be in the millimolar range (*Manson et al., 1985*; *Silhavy et al., 1975*), which is comparable to the concentration of SBDs in the OpuA transport complex. The transient and weak interactions between SBDs offer a new perspective on how transport efficiency of proteins can be regulated.

## Materials and methods

**Key resources table**

| Reagent type (species) or resource | Designation | Source or reference | Identifiers | Additional information |
|---|---|---|---|---|
| Gene (*Lactococcus lactis* IL1403) | OpuAA | NA | Q9RQ05 | |
| Gene (*Lactococcus lactis* IL1403) | OpuABC | NA | Q9KIF7 | |
| Strain, strain background (*Lactococcus lactis*) | Opu401 | 10.1073/pnas.0603871103; *Biemans-Oldehinkel et al., 2006* | | *Lactococcus lactis* NZ9000 with OpuA gene deleted |
| Strain, strain background (*Escherichia coli*) | BL21(DE3) | NA | | |

*Continued on next page*

*Continued*

| Reagent type (species) or resource | Designation | Source or reference | Identifiers | Additional information |
|---|---|---|---|---|
| Recombinant DNA reagent | pNZopuAhis | 10.1073/pnas.97.13.7102; *van der Heide and Poolman, 2000* | | Expression plasmid for OpuA |
| Recombinant DNA reagent | pMSP1E3D1 | Addgene | CAT#:20066 | Expression plasmid for MSP1E3D1 |
| Commercial assay or kit | Pierce BCA Protein Assay Kit | ThermoFisher Scientific Inc. | CAT#:23225 | |
| Chemical compound, drug | Alexa Fluor 555 | ThermoFisher Scientific Inc. | CAT#: A20346 | |
| Chemical compound, drug | Alexa Fluor 647 | ThermoFisher Scientific Inc. | CAT#: A20347 | |
| Chemical compound, drug | 1,2-dioleoyl-sn-glycero-3-phosphoethanolamine (*DOPE*) | Avanti Polar Lipids Inc. | CAT#: 850725 P | |
| Chemical compound, drug | 1,2-dioleoyl-sn-glycero-3-phosphocholine (DOPC) | Avanti Polar Lipids Inc. | CAT#: 850375 P | |
| Chemical compound, drug | 1,2-dioleoyl-sn-glycero-3-phospho-(1'-rac-glycerol) (DOPG) | Avanti Polar Lipids Inc. | CAT#: 840475 P | |
| Chemical compound, drug | *n*-Dodecyl-β-d-maltoside (*DDM*) | Glycon Biochemicals GmbH | CAT#: D97002 | |
| Chemical compound, drug | Adenosine 5'-triphosphate (ATP) | Roche Holding AG | CAT#: 10519987001 | |
| chemical compound, drug | ß-Nicotine amide adenine dinucleotide (NADH) | Carl Roth Gmbh | CAT#: AE12.1 | |
| Chemical compound, drug | Phospho*enol*pyruvic acid (PEP) | Carl Roth Gmbh | CAT#: 8397.3 | |
| Chemical compound, drug | Pyruvate Kinase/Lactic Dehydrogenase enzymes from rabbit muscle (PK/LDH) | Sigma-Aldrich | CAT#: P0294 | |
| Chemical compound, drug | Glycine betaine | Sigma-Aldrich | CAT#: 61962 | |
| Chemical compound, drug | L-glutamine | Sigma-Aldrich | CAT#: G3126 | |
| Chemical compound, drug | L-arginine | Sigma-Aldrich | CAT#: A5006 | |
| Chemical compound, drug | *Ortho*vanadate | Sigma-Aldrich | CAT#: 450243 | |
| Software, algorithm | FRETbursts | 10.1371/journal.pone.0160716; *Ingargiola et al., 2016b* | | Version 0.7.1 |
| Software, algorithm | Bursth2m | 10.1038/s41467-022-28632-x; *Harris et al., 2022* | | Version 0.1.6 |
| Software, algorithm | Phconvert | phconvert.rtfd.io; *Ingargiola et al., 2016a* | | Version 0.9 |
| Software, algorithm | Ggplot2 | 10.1007/978-3-319-24277-4; *Wickham, 2016* | | Version 3.3.5 |
| Software, algorithm | SymPhoTime 64 | PicoQuant | RRID:SCR_016263 | Version 2.6 |
| Software, algorithm | SerialEM | 10.1016/j.jsb.2005.07.007; *Mastronarde, 2005* | | Version 4.0.10 |
| Software, algorithm | FOCUS | 10.1016/J.JSB.2017.03.007; *Biyani et al., 2017* | | Version 1.0.0 |
| Software, algorithm | MotionCor2 | 10.1038/nmeth.4193; *Zheng et al., 2017* | | |

*Continued on next page*

*Continued*

| Reagent type (species) or resource | Designation | Source or reference | Identifiers | Additional information |
|---|---|---|---|---|
| Software, algorithm | CTFFIND4 | 10.1016/J.JSB.2015.08.008; **Rohou and Grigorieff, 2015** | | |
| Software, algorithm | crYOLO | 10.1038/s42003-019-0437-z; **Wagner et al., 2019** | | Version 1.8.4 |
| Software, algorithm | Relion | 10.1042/BCJ20210708; **Kimanius et al., 2021** | | Version 4.0 |
| Software, algorithm | cryoSPARC | 10.1038/nmeth.4169; **Punjani et al., 2017** | | Version 4.1.1 |
| Software, algorithm | UCSF ChimeraX | 10.1002/pro.3235; **Goddard et al., 2018** | | Version 1.5 |

## Materials

All the oligonucleotide primers are listed in *Table 5*. The OpuA mutations were introduced in pNZopuAhis by means of USER cloning (*van der Heide and Poolman, 2000*). This plasmid contains the *opuA* genes from *L. lactis* IL1403 with a C-terminal His$_6$-tag on the *opuABC* gene. The ultracentrifuge rotors were from Beckman Coulter, Brea, California.

The general purification buffer for OpuA purification in detergent consisted of 50 mM KPi pH 7.0, 200 mM KCl, 20% glycerol (v/v) plus 0.04% DDM (w/w) (Buffer 1). The general purification buffer for OpuA purification in nanodiscs consisted of 20 mM K-HEPES pH 7.0 plus 300 mM KCl (Buffer 2). During the different purification steps, the buffers were supplemented with variable concentrations of imidazole pH 7.5 as described below. In case of the cysteine variants of OpuA, 1 mM DTT was added to each step of the protocol from the breaking of the cells and onwards.

A detailed purification and nanodisc reconstitution procedure for OpuA has been made publicly available at protocol.io (https://doi.org/10.17504/protocols.io.bp2l624pzgqe/v1). This protocol also encompasses details regarding the optimized parameters of the procedure.

## Expression and purification of OpuA

OpuA variants were expressed as described before (*Sikkema et al., 2020*). The cells with an optical density at 600 nm (OD$_{600}$) of less than 200 were stored at –80 °C in 50 mM KPi pH 7.5 with 20% glycerol (v/v).

The cells were thawed and supplemented with 100 µg/mL DNAse, 2 mM MgSO$_4$ and 1 mM PMSF while stirring, and broken at 30 kPsi, using a high pressure lyser HPL6 (Maximator GmbH, Nordhausen, Germany). Afterwards, 5 mM EDTA pH 8.0 was added and cell debris was removed by centrifugation (20 min, 15,000 x *g* 4 °C). Crude membrane vesicles (CMVs) were spun down by ultracentrifugation in a Type 45-Ti rotor (135 min, 138,000 x *g*, 4 °C), resuspended in half of the original volume in 50 mM KPi pH 7.5 with 20% glycerol (v/v) and spun down again in a Type 50.2-Ti rotor (75 min, 185,000 x

**Table 5.** Primers used in this study to generate mutations in OpuA.

| Oligo nr. | Oligo name | Sequence (from 5' to 3') * |
|---|---|---|
| 9053 | UOpuAC-K521C fw | aaaggttUgaagTGTgaaaatccagaagcttataaag |
| 9054 | UOpuAC-K521C rev | aaaccttUacgaacaatgg |
| 9511 | UOpuAC-N414C fw | aattgaagaUttaacaaatcaagc |
| 9512 | UOpuAC-N414C rev | atcttcaatUgaattaacACAcatataacttggaac |
| 9617 | UOpuAA-E190Q fw | aagctttcUctgctcttgac |
| 9618 | UOpuAA-E190Q rev | agaaagcTUGatccatgagcaaaatc |
| 9619 | UOpuAC-V149Q fw | atttgatUccagcgCAAgcattctttgg |
| 9620 | UOpuAC-V149Q rev | aatcaaaUaaacgaaaccagg |

*Mutated triplets and introduced uracils are in capitals.

*g*, 4 °C). The washed CMVs were resuspended in the same buffer to a concentration of 8–16 mg/mL of total protein, flash frozen in liquid nitrogen and stored at –80 °C. Total protein concentration was determined by a Pierce BCA Protein Assay Kit (ThermoFisher Scientific Inc, Waltham, Massachusetts).

For solubilization the CMVs were thawed and incubated in an open MLA-80 centrifuge tube for 30 min at 4 °C in 6 mL Buffer 1, supplemented with 0.5% DDM (w/w), at a total protein concentration of 3 mg/mL. The sample was mixed once after 15 min by gently pipetting. Insolubilized material was spun down by ultracentrifugation (20 min, 337,000 x *g*, 4 °C) and 10 mM imidazole pH 7.5 was added to the supernatant. The supernatant was poured into a column with a closed outlet that was loaded with 0.5 mL Ni$^{2+}$-Sepharose resin (GE Healthcare, Chicago, Illinois), which was pre-washed two times with 6 column volumes of MQ and equilibrated with 4 column volumes of Buffer 1, supplemented with 10 mM imidazole pH 7.5. The solution was mixed by gently pipetting. After 10–20 min, the solution was mixed again by pipetting. Once a part of the resin was sedimented, the column outlet was opened to let the protein solution flow through. The resin was washed with two times 10 column volumes of Buffer 1, supplemented with 50 mM imidazole pH 7.5. OpuA was eluted with Buffer 1, supplemented with 200 mM imidazole pH 7.5, in a first fraction of 0.6 column volume and four subsequent fractions of 0.4 column volume. Purified OpuA was directly used for reconstitution in nanodiscs.

## Expression and purification of MSP1E3D1 without His-tag

The expression and purification method of MSP1E3D1 was adapted from a previously described method (*Ritchie et al., 2009*). An *E. coli* BL21(DE3) colony, transformed with pMSP1E3D1, was grown during the day in 2 mL Luria–Bertani (LB) medium (1% bactotrypton, 0.5% yeast extract, 1% NaCl)+30 µg/mL kanamycin at 37 °C and 200 rpm. A total of 500 µL of the culture was used to make a dilution series of four samples for overnight growth at 37 °C and 200 rpm with 50 mL LB +30 µg/mL kanamycin, with dilution factors of 100 between the samples. The sample with the highest overall dilution and having visible growth was used as starter culture for 2 L of phosphate-buffered Terrific Broth medium, 200 µL antifoam 204 plus 10 µg/mL kanamycin. The bioreactor (2 L) temperature was set at 37 °C, and the mixing speed was set at 500 rpm with a linear increase to 1000 rpm in 4 hr. Aeration was provided by a filtered airflow of 40 L/min through the media. Expression was induced with 1 mM IPTG at an OD$_{600}$ of 2. After 3 hr, the cells were harvested at 6000 x *g* for 15 min at 4 °C, washed in 50 mM KPi pH 7.8, and finally resuspended in 50 mM KPi pH 7.8 in less than 10% of the original volume, flash frozen in liquid nitrogen, and stored at –80 °C. Cells from one bioreactor were enough for two separate purifications.

The cells were thawed and supplemented with 100 µg/mL DNase, 1 mM phenylmethylsulfonyl fluoride, and 1% (v/v) TritonX-100. MSP1E3D1 was released from the cytosol by sonication in an ice-ethanol bath for 72 cycles of 5 s with a 5 s interval and with an amplitude of 56 µm. Cell debris and whole cells were removed by centrifugation (30 min, 30,000 x *g*, 4 °C). 20 mM imidazole pH 7.5 was added to the supernatant.

For purification of MSP1E3D1, 7.5 mL of Ni$^{2+}$-Sepharose resin was used, which was washed with 4 column volumes demi water and 4 column volumes 50 mM KPi pH 7.8. The supernatant was incubated with the resin plus 20 mM imidazole for 1 hr at 4 °C under gentle agitation. After flowthrough, the column was washed in three steps, using three different wash buffers of each five column volumes. Each wash buffer contained 40 mM Tris-HCl pH 8.0 plus 300 mM NaCl. The wash buffer was supplemented with 1% Triton X-100 (1), 50 mM Na-cholate plus 20 mM imidazole pH 8.0 (2), or 50 mM imidazole pH 8.0 (3). The protein was eluted in twelve 2 mL fractions of 40 mM Tris-HCl pH 8.0, 300 mM NaCl plus 500 mM imidazole pH 8.0 and incubated under gentle agitation for 90 min at 4 °C with His-tagged TEV-protease (weight-based MSP:TEV ratio of 40:1) plus 5 mM EDTA to remove the His-tag of MSP1E3D1. The sample was dialyzed overnight against 20 mM tris-HCl pH 7.4, 100 mM NaCl, 0.5 mM DTT plus 0.5 mM EDTA at 4 °C while the dialysis buffer was gently stirred.

After dialysis, 2.5 mL of Ni$^{2+}$-Sepharose resin was used to remove the TEV-protease and noncleaved MSP1E3D1. The resin was washed with 6 column volumes of demi water and 6 column volumes of 20 mM Tris-HCl pH 7.4, 100 mM NaCl. The flow-through was reapplied twice to make sure all TEV-protease and noncleaved MSP1E3D1 was bound to the resin. The third time, the flowthrough was collected as a single fraction. Subsequently, the column was washed with 20 mM Tris-HCl pH 7.4, 100 mM NaCl plus 40 mM imidazole in nine 2 mL fraction. All MSP1E3D1 containing fractions were

pooled and concentrated to 150–175 µM MSP1E3D1, using a Vivaspin 20 centrifugation filter (10 kDa cutoff). Aliquots were flash frozen in liquid nitrogen and stored at –80 °C.

## Preparation synthetic lipids stock

The three lipid species DOPE, DOPC, DOPG were separately dissolved in chloroform to a concentration of 25 mg/mL and mixed in a volume ratio of 50:12:38 (DOPE:DOPC:DOPG). The lipids were dried in a rotary evaporator at 40 °C and ~450 mbar for 20 min, followed by 30 min at full vacuum. The lipids were then resuspended in diethyl ether and subsequently dried in a rotary evaporator at 40 °C and ~900 mbar for 20 min, followed again by 30 min at full vacuum. Residual diethyl ether was removed by applying a gentle $N_2$ flow to the lipid film for 5 min. The lipids were resuspended in 50 mM KPi pH 7.0 to a concentration of 25 mg/mL and sonicated in an ice-water bath for 16 cycles of 15 s with a 45 s interval and with an amplitude of 77 µm. Aliquots of the lipids were fast frozen in liquid nitrogen and slowly thawed at room temperature in three cycles. Subsequently, the aliquots were stored in liquid nitrogen.

## Reconstitution of OpuA in nanodiscs

Lipids were thawed, diluted four times in 50 mM KPi pH 7.0 and sonicated in an ice-water bath for 8 cycles of 15 s with a 45 s interval at an amplitude of 77 µm. Afterwards, 1% DDM (w/w) was added from a 10% stock to fully solubilize the lipids at a concentration of 5.63 mg/mL lipids. The solution was left at RT for at least 2 hr before it was used.

For optimal reconstitution, 0.9 mM lipids, 45 µM MSP1E3D1 plus 4.5 µM OpuA (based on 218,162 Da molecular weight) were incubated in 50 mM KPi pH 7.0, and in a volume of 700–1000 µL. This was done in 1.5 mL Eppendorf tubes to minimize the hazardous air-water interface. The concentration of OpuA after IMAC was generally around 18 µM but always higher than 13 µM. This is necessary, because OpuA is eluted in the presence of 20% (v/v) glycerol. It was aimed to dilute OpuA enough in the reconstitution mixture to get the glycerol content during reconstitution below 7% (v/v) and hence minimize the negative effects of glycerol on the reconstitution efficiency. The lipids were diluted out more than 8 times to reach a concentration of 0.9 mM lipids. This also reduced the total DDM concentration in the reconstitution mixture to below 8 mM, which is necessary to prevent dissociation of OpuAA from the OpuA complex. After gentile agitation at 4 °C for 1 hr, the sample was incubated overnight with 500 mg SM-2 Bio-Beads (Bio-Rad, Hercules, California) under the same conditions.

The next day, the nanodiscs were separated from the Bio-Beads by means of a syringe with a needle. Aggregates were removed by centrifugation (15 min, 20,000 x *g*, 4 °C). The sample was loaded to 0.2 mL $Ni^{2+}$-Sepharose resin, which was washed two times with 5 column volumes of MQ and equilibrated with 5 column volumes of 50 mM KPi pH 7.0. The flowthrough was reapplied twice, before the resin was washed two times with 5 column volumes of Buffer 2, supplemented with 25 mM imidazole pH 7.5. OpuA was eluted with Buffer 2, supplemented with 200 mM imidazole pH 7.5, in a first fraction of 0.8 column volume and a second fraction of 2.5 column volumes. The second fraction was directly further purified by size exclusion chromatography using a Superdex 200 increase 10/300 GL column in Buffer 2. The purified protein was flash frozen in liquid nitrogen and stored at –80 °C.

## ATPase assays

The ATPase activity assays were performed as is described in *Sikkema et al., 2020*. In this coupled enzyme assay, the synthesis of ATP from ADP is coupled to oxidation of NADH to $NAD^+$, which can be monitored over time by measuring NADH-based absorbance at 340 nm. In other words, for each ATP hydrolysis reaction, one NADH molecule gets dehydrogenated.

In short, each measured sample contained 50 mM K-HEPES pH 7.0, 4 mM phospho*enol*pyruvate, 600 µM NADH, 2.1–3.5 U of pyruvate kinase, 3.2–4.9 U of lactate dehydrogenase plus approximately 5 µg/mL OpuA in nanodiscs. The concentrations of glycine betaine, KCl, and Mg-ATP were varied as described in the Results section (*Figure 1—figure supplement 1*, *Figure 3*). Absolute activities in $min^{-1}$ *are based on a m*olecular weight of two MSP1E3D1 and one full-length OpuA complex (total mass 278,126 Da). The data was analyzed in R, using ggplot2 (*Wickham, 2016*).

## Labeling of OpuA for smFRET experiments

For the stochastic labeling, only half of the reconstituted nanodisc sample was used (i.e. ~3 nmol of cysteines). Therefore, the sample was loaded to 0.1 mL Ni$^{2+}$-Sepharose resin, which had been washed and equilibrated as described under 'Reconstitution of OpuA in nanodiscs'. The flow through was reapplied twice, before the resin was washed twice with 5 column volumes of Buffer 2 to remove the DTT. Directly after the wash steps 4 column volumes of Buffer 2, supplemented with 50 nmol of Alexa Fluor 555 plus 50 nmol of Alexa Fluor 647, were added to the resin. The flow was stopped by closing the outlet when more than 2 column volumes were flown through, but before the column ran dry.

The column was wrapped with parafilm and aluminium foil, and incubated on ice for 3–4 hr. Afterwards, the resin was washed twice with 10 column volumes of Buffer 2, supplemented with 25 mM imidazole pH 7.5. OpuA was eluted with Buffer 2, supplemented with 300 mM imidazole pH 7.5, in a single fraction of 4 column volumes, which was directly further purified by size exclusion chromatography, using a Superdex 200 increase 10/300 GL column in Buffer 2. The sample after size-exclusion chromatography was aliquoted, flash frozen in liquid nitrogen and stored at –80 °C.

The efficiency of labeling with Alexa fluorophores was determined before aliquoting and storage, by means of an absorbance scan using a UV-VIS spectrophotometer (Cary 100 Bio; Varian Inc, Palo Alto, California).

## smFRET measurements

Pulsed Interleaved Excitation (PIE) and solution-based smFRET experiments were performed on a MicroTime 200 confocal microscope (PicoQuant, Berlin, Germany). Prior to the recording, microscope slides (170 $\mu$m thickness, No. 1.5 H precision cover slides, VWR Marienfeld, Leicestershire, Great Britain; LH26.1) were coated for at least one min with 1 mg/mL filtered (0.2 $\mu$m) bovine serum albumin (BSA) in 50 mM HEPES-K pH 7.0, after which the BSA solution was removed by pipetting and replaced by 150–200 $\mu$L of the sample.

The laser pulse rate was set at 40 MHz. Fluorophores were alternately excited, using a 532 nm (LDH-P-FA-530-B; PicoQuant, Berlin, Germany) and 638 nm (LDH-D-C-640; PicoQuant, Berlin, Germany) laser. The laser beam was focused 7 $\mu$m away from the glass-solution interface in the $z$-direction, by means of an oil-immersed objective lens (UPlanSApo 100x1.40 NA; Olympus, Tokyo, Japan). The emitted photons from the sample were coordinated through a 100 $\mu$m pinhole, separated through a laser beam-splitter (ZT640RDC; Chroma Technology, Bellows Falls, Vermont), filtered by either a HQ690/70 (Chroma Technology, Bellows Falls, Vermont) or a 582/75 (Semrock, Rochester, New York) emission filter, and recorded by two photon counting modules (donor photons: SPCM-AQRH-14-TR, acceptor photons: SPCM-CD-3516-H; Excelitas Technologies, Waltham, Massachusetts).

## smFRET data analysis

Raw data were saved as.PTU file extensions by the SymPhoTime 64 software (PicoQuant, Berlin, Germany), but was converted to.HDF5 file extensions, using the Python package phconvert (*Ingargiola et al., 2016a*). Fluorescent bursts were identified by the all photon burst search (APBS) method, using the Python based software package FRETbursts (*Ingargiola et al., 2016b*). The Python codes for the downstream analysis were adapted from published codes (*Harris et al., 2022*). Burst cut-off settings were set to a minimum of m=10 consecutive photons per sliding window and at least $F$=6 times higher than the background signal. The background signal was calculated over each 30 s of the measurement. The initial burst selection considered all detected photons in the donor and acceptor channel during both the donor and acceptor excitation periods. This was done with a cut-off of minimally 35 photons per burst.

Three types of photon counts were extracted from each burst: donor-based donor emission ($F_{DD}$), donor-based acceptor emission ($F_{DA}$) and acceptor-based acceptor emission ($F_{AA}$). These were used to calculate the apparent FRET efficiency E* and stoichiometry S of each burst. Apparent FRET efficiency was defined as:

$$E^* = \frac{F_{DA}}{F_{DA} + F_{DD}} \tag{1}$$

The stoichiometry of a burst was defined as the ratio between overall donor-based fluorescence over the total fluorescence of the burst, that is:

$$S = \frac{F_{DA} + F_{DD}}{F_{DA} + F_{DD} + F_{AA}} \tag{2}$$

For mpH2MM, the Python package bursth2mm was used and run with default parameters. All calculations can be found back in the supplemented Jupyter notebooks. The model optimization process for each model is based on the ICL criterion. This criterion takes into account the probability of the most likely state path through the bursts and penalizes for the number of states that is needed to explain the data (*Harris et al., 2022*). The model with lowest ICL was adopted for downstream analysis. For each calculation, also the BIC' value was inspected, which was usually slightly above the 0.05 cut-off for the optimal model, based on the ICL criterion.

The $\gamma$-correction factor was calculated on the basis of the resulting two FRET states after mpH2MM modeling, and applying the leakage and crosstalk corrections to the dwells (*Hohlbein et al., 2014*). Subsequently, a third burst selection with the same cut-off settings was performed on the initial data, but now by applying the three correction factors. The bursts were further analyzed by mpH2MM, as described above. The resulting dwells were corrected by applying the three correction factors again.

The bursts were also analyzed by BVA (*Torella et al., 2011*). In a BVA, the mean FRET efficiency of each burst is plotted against its standard deviation ($\sigma$) from the mean during the burst. The data was binned with a bin size of 0.1 E and plotted in the same graph. The $\sigma$ of the binned data was compared with what would be expected from theory. Deviation from the expected $\sigma$ was considered as a qualitative indication for within-burst dynamics.

## Steady-state fluorescence anisotropy measurements

To measure polarized fluorescence, a Jasco FP-8300 scanning spectrofluorometer (Jasco Inc, Easton, Maryland) with polarized filters was used. Alexa Fluor 555 and Alexa Fluor 647 were excited at 535 nm and 635 nm, respectively (10 nm bandwidth). Emission was recorded at 580 nm and 660 nm (10 nm bandwidth). Anisotropy (r) was calculated by:

$$r = \frac{I_{VV} - G * I_{VH}}{I_{VV} + 2G * I_{VH}} \tag{3}$$

where $I_{VV}$ and $I_{VH}$ represent fluorescence intensities after vertical excitation in the vertical and horizontal plane, respectively. Parameter *G* is a correction factor to compensate for different sensitivity of the machine in the vertical versus the horizontal plane:

$$G = \frac{I_{HV}}{I_{HH}} \tag{4}$$

## Sample preparation for single particle cryo-EM and data acquisition

Freshly purified wild type OpuA in MSP1E3D1 nanodiscs was concentrated to 1 mg/ml with a Vivaspin 500 (100 kDA). A concentrated sample (2.8 $\mu$l) was applied to holey carbon grids (Au R1.2/1.3, 300 mesh, Quantifoil, Jena, Germany) previously glow-discharged for 30 s at 5 mA. The grids were blotted for 3–5 s at 15–22°C and 100% humidity in a Vitrobot Mark IV (ThermoFisher Scientific Inc, Waltham, Massachusetts), subsequently plunge-frozen into a liquid ethane/propane mixture, and stored in liquid nitrogen for data collection.

Datasets for each condition were collected in-house using a 200 kV Talos Arctica microscope (ThermoFisher Scientific Inc, Waltham, Massachusetts). Movies were recorded with a K2 summit (Gatan Inc, Pleasanton, California) with a post-column BioQuantum energy filter (Gatan Inc, Pleasanton, California) with a zero-loss slit width of 20 eV. Automatic data collection was performed using SerialEM ver 4.0.10 with a 100 $\mu$m objective aperture, a pixel size of 1.022 Å (calibrated magnification 48,924), a defocus range of –0.5 to –2.0 μm, a total exposure time of 9 s captured in 60 frames (150 ms subframe exposure) and total electron exposure of 50.9 e⁻ per Å² (*Mastronarde, 2005*; *Schorb et al., 2019*). Target holes for data acquisition were qualitatively screened and selected based on their ice thickness (20–50 nm) with an in-house-developed script (*Rheinberger et al., 2021*). Data quality was monitored on-the-fly using FOCUS version 1.0.0 (*Biyani et al., 2017*).

## Cryo-EM image processing of wild type OpuA in 50 mM KCl

A total of 5462 movies were collected from five grids (containing samples from the same purification and frozen in the same session) and were gain corrected and pre-processed on-the-fly with FOCUS version 1.0.0 utilizing MotionCor2 and CTFFIND4 for motion correction (frame-dose weighting) and CTF estimation, respectively (*Zheng et al., 2017*; *Rohou and Grigorieff, 2015*). In total, 4482 images were selected for further processing after removing those of lower quality (visual inspection, poor CTF estimation and contamination). A total of 1,879,237 particles were picked utilizing the PhosaurusNet architecture of crYOLO version 1.8.4 with the general model and subsequently extracted in Relion 4.0 with a box size of 256 pixels and downscaled to 2.044 Å/pixel (*Wagner et al., 2019*; *Kimanius et al., 2021*). Extracted particles were processed in parallel both in Relion 4.0 and CryoSPARC 4.1.1.

In CryoSPARC, particles were subjected to two rounds of 2D classification using default settings with the exception of 'Number of final full iterations' set to 5 and an 'Initial classification uncertainty factor' of 1 and 2 for the first round and the second round, respectively. Selected particles (552,280) were subjected to *ab initio* reconstruction and subsequently to hetero refinement. The best class (166,066 particles) was further refined using non-uniform refinement which resulted in 6.21 Å resolution. The final map was low pass filtered to 15 Å for comparison. All processing was done without imposing any symmetry (C1).

## Cryo-EM image processing of wild type OpuA in 100 mM KCl

A total of 1,474 movies were collected from a single grid and were gain corrected and pre-processed on-the-fly with FOCUS version 1.0.0 utilizing MotionCor2 and CTFFIND4 for motion correction (frame-dose weighting) and CTF estimation, respectively. In total, 1051 images were selected for further processing after removing those of lower quality (visual inspection, poor CTF estimation and contamination). A total of 559,491 particles were picked utilizing the PhosaurusNet architecture of crYOLO version 1.8.4 with a pretrained model as described in *Sikkema et al., 2020* and subsequently extracted in Relion 4.0 with a box size of 256 pixels and downscaled to 2.044 Å/pixel. Extracted particles were subjected to a round of 2D classifications with a mask of 200 Å. Selected particles (264,267) were classified in 3D by limiting the E-step size to 7 Å, regularization parameter T=4 and a mask of 200 Å. The best three classes (97,369 particles) were selected and re-extracted with a box size of 256 pixels with no downscaling. These particles were subsequently imported into CryoSPARC and subjected to *ab initio* reconstruction and subsequently to hetero refinement. The best classes (80,348 particles) were further refined using non-uniform refinement which resulted in 7 Å resolution. The final map was low-pass filtered to 15 Å for comparison. All processing was done without imposing any symmetry (C1).

## Acknowledgements

This work was financed by the NWO Gravitation Program BaSyC. CP acknowledges funding from the Dutch Research Council: Nederlandse Organisatie voor Wetenschappelijk Onderzoek (NWO) Veni Grant 722.017.001, NWO Start-Up Grant 40.018.016 and together with BP the NWO OCENW. KLEIN.526.

## Additional information

### Funding

| Funder | Grant reference number | Author |
| --- | --- | --- |
| Nederlandse Organisatie voor Wetenschappelijk Onderzoek | Gravitation Program BaSyC | Bert Poolman |
| Nederlandse Organisatie voor Wetenschappelijk Onderzoek | 722.017.001 | Cristina Paulino |

| Funder | Grant reference number | Author |
| --- | --- | --- |
| Nederlandse Organisatie voor Wetenschappelijk Onderzoek | 40.018.016 | Cristina Paulino |
| Nederlandse Organisatie voor Wetenschappelijk Onderzoek | OCENW.KLEIN.526 | Cristina Paulino Bert Poolman |

The funders had no role in study design, data collection and interpretation, or the decision to submit the work for publication.

### Author contributions
Marco van den Noort, Conceptualization, Data curation, Formal analysis, Investigation, Methodology, Writing - original draft, Writing - review and editing; Panagiotis Drougkas, Data curation, Formal analysis, Investigation, Methodology, Writing - original draft, Writing - review and editing; Cristina Paulino, Bert Poolman, Conceptualization, Funding acquisition, Writing - review and editing

### Author ORCIDs
Marco van den Noort ⓘ http://orcid.org/0000-0002-2025-461X
Bert Poolman ⓘ http://orcid.org/0000-0002-1455-531X

Reviewer #1 (Public Review): https://doi.org/10.7554/eLife.90996.3.sa1
Reviewer #2 (Public Review): https://doi.org/10.7554/eLife.90996.3.sa2
Author response https://doi.org/10.7554/eLife.90996.3.sa3

---

# Additional files

### Supplementary files
• MDAR checklist

### Data availability
All raw single photon data together with the Python scripts to analyze the data, plus the cyro-EM maps are deposited in the online database DataverseNL.

The following dataset was generated:

| Author(s) | Year | Dataset title | Dataset URL | Database and Identifier |
| --- | --- | --- | --- | --- |
| Van den Noort M, Drougkas P, Paulino O, Poolman D | 2023 | Raw smFRET data, analysis scripts and cryo-EM maps for: The substrate-binding domains of the osmoregulatory ABC importer OpuA physically interact | https://doi.org/10.34894/GSIEBW | DataverseNL, 10.34894/GSIEBW |

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
