## [Editor Report · eLife assessment]

The OpuA Type I ABC importer uses two substrate binding domains to capture extracellular glycine betaine and present the substrate to the transmembrane domain for subsequent transport and correction of internal dehydration. This study presents **valuable** findings addressing the question of whether the two substrate binding domains of OpuA dock and physically interact in a salt-dependent manner. The single-molecule fluorescence resonance energy transfer and cryogenic electron microscopy data that are presented provide **convincing** support for the existence of a transient interaction between the substrate binding domains that depends on ionic strength, laying a foundation for future studies exploring how this interaction is involved in the overall transport mechanism.

---

## [Referee Report · Reviewer #1 (Public Review)]

Summary: The type I ABC importer OpuA from Lactococcus lactis is the best studied transporter involved in osmoprotection. In contrast to most ABC import systems, the substrate binding protein is fused via a short linker to the transmembrane domain of the transporter. Consequently, this moiety is called the substrate binding domain (SBD). OpuA has been studied in the past in great detail and we have a very detailed knowledge about function, mechanisms of activation and deactivation as well as structure.

Strengths: Application of smFRET to unravel transient interactions of the SBDs. The method is applied at a superb quality and the data evaluation is excellent.

Weaknesses: The proposed model is not directly supported by experimental data. Rather alternative models are excluded as they do not fit to the obtained data. However, this is now clearly stated in the manuscript

---

## [Referee Report · Reviewer #2 (Public Review)]

Summary:

In this report the authors used solution-based single-molecule FRET and low resolution cryo-EM to investigate the interactions between the substrate-binding domains of the ABC-importer OpuA from Lactococcus lactis. Based on their results, the authors suggest that the SBDs interact in an ionic strength-dependent manner.

Strengths:

The strength of this manuscript is the uniqueness and importance of the scientific question, the adequacy of the experimental system (OpuA), and the combination of two very powerful and demanding experimental approaches.

Weaknesses:

A demonstration that the SBDs physically interact with one another, and that this interaction is important for the transport mechanism will greatly strengthen the claims of the authors. The relation to cooperativity is also unclear.

---

## [Author Response]

The following is the authors’ response to the original reviews.

**Recommendations for the authors**

**Reviewer #1 (Recommendations For The Authors):**
Below I summarize points that should be addressed in a revised version of the manuscript.Page 6, first paragraph: I don't understand by the signals average out to a single state. If the distribution is indeed randomly distributed, a broad signal with low intensity should be present.

We agree that this statement may cause confusion. We changed the text (marked in bold) to clarify the statement: The mobility of the undocked SBDs will be higher than the diffusion of the whole complex, allowing the sampling of varying interdomain distances within a single burst. However, these dynamic variations are subsequently averaged to a singular FRET value during FRET calculations for each burst, and may appear as a single low FRET state in the histograms.

Page 6, third paragraph: how can the donor only be detected in the acceptor channel? Is this tailing out?

Donor only signal is not detected in the acceptor channel. As described in page 5 and in the Materials & Methods section, the dye stoichiometry value is defined for each burst/dwell using three types of photon counts: donor-based donor emission (FDD), donor-based acceptor emission (FDA) and acceptorbased acceptor emission (FAA).S=FDA+FDDFDA+FDD+FAA

When no acceptor fluorophore is present FAA=0 and S=1.

Some donor photons bleed through into the acceptor channel, but we correct for this by calculating the leakage and crosstalk factors as described in the Materials and Methods (page 20).

We changed the text (marked in bold) in the manuscript to address the question: The FRET data of both OpuA variants is best explained by a four-state model (Figure 2A,B; fourth and fifth panel) (Supplementary File 3). Two of the four states represent donor-only (S≈1) or acceptor-only (S≈0) dwells. The full bursts belonging to donor-only and acceptor-only molecules were excluded prior to mpH2MM. This means that some molecules transit to a donor-only or acceptor-only state within the burst period, which most likely reflects blinking or bleaching of one of the fluorophores. These donoronly and acceptor-only states were also excluded during further analysis. The other two states reflect genuine FRET dwells that were analyzed by mpH2MM. They represent different conformations of the SBDs.

Page 7, "SBD dynamics ..": why was the V149Q mutant only analyzed in the K521C background and not also in the N414C background?

The two FRET states were best distinguished in OpuA-K521C. Therefore, we decided to focus on OpuA-K521C and not OpuA-N414C. OpuA-V149Q was used to show that reduced docking efficiency does not affect the transition rate constants and relative abundances of the two FRET states, and we regarded it sufficient to test the SBD dynamics in OpuA-K521C only.

Page 8, second paragraph: why was the N414C mutant analyzed only from 0 - 600 mM and not also up to 1000 mM?

In line with the previous answer, our main focus was on OpuA-K521C, since the two FRET states were best distinguished in OpuA-K521C. OpuA-N414C was used to prove that similar states are observed when measuring with fluorophores on the opposite site of the SBD. We studied how the FRET states change in response to different conditions that correspond to different stages of the transport cycle and how it changes in response to different ionic strengths. Initially, 600 mM KCl was used to study the dynamics of the SBD at high ionic strength. Later in this study, we tested a very wide range of different salt concentrations for OpuA-K521C to get detailed insights into the dynamics of the SBDs over a wide ionic strength range. Note that 1 M KCl is a very high, non-physiological ionic strength for the typical habitat of L. lactis and was only used to show that the high FRET state occurs even under very extreme conditions.

Page 8, third paragraph: why was the dimer (if it is the source of the FRET signal) only partially disrupted?

We acknowledge that this is a very good point. However, we purposely did not speculate on this point in the manuscript, because we have limited information on the molecular details of the interaction. As we highlight on page 8, the SBDs experience each other in a very high apparent concentration (millimolar range). This means that the interactions are most likely very weak (low affinity) and not very specific. Such interactions are in the literature referred to as the quinary structure of proteins and they occur at the high macromolecular crowding in the cell and in proteins with tethered domains, and thus at high local concentrations. Such interactions can be screened by high ionic strength. In the revised manuscript, we now present the partially disrupted dimer structure in the context of the quinary structure of a protein (page 11):

In other words, the high FRET state may comprise an ensemble of weakly interacting states rather than a singular stable conformation, resembling the quinary structure of proteins. The quinary structure of proteins is typically revealed in highly crowded cellular environments and describes the weak interactions between protein surfaces that contribute to their stability, function, and spatial organization (Guin & Gruebele, 2019). Despite the current study being conducted under dilute conditions, the local concentration of SBDs (~4 mM) mimics a densely populated environment and reveal quinary structure.

Page 9, second paragraph: according to the EM data processing, only 20% of the particles were used for 3D reconstruction. Why? Does it mean that the remaining 80% were physiologically not relevant? If so, why were the 20% used relevant?

We note that it is a fundamental part of image processing of single particle cryo-EM data to remove false positives or low-resolution particles throughout the processing workflow. In particular when using a very low and therefore generous threshold during automated particle picking, as we did (t=0.01 and t=0.05 for the 50 mM KCl and 100 mM KCl datasets, respectively), the initial set of particles includes a significant amount of false positives – a tradeoff to avoid excluding particles belonging to low populated classes/orientations. It is thus common that more than 50% of ‘particles’ are excluded in the first rounds of 2D classification. In our case, only 30% and 52% of particles were retained after such first clean-up steps. Subsequently, the particle set is further refined, and additional false positives and low-resolution particles are excluded during extensive rounds of 3D classification. We also note that during the final steps, most of the data excluded represents particles of lower quality that do not contribute to a high-resolution, or belong to low population protein conformations. This does not mean that such a population is not physiological relevant. In conclusion, having only 5-20% of the initial automated picked particles contributing to the reconstruction of the final cryo-EM map is common, with the vast majority of excluded particles being false positives.

Page 11, third paragraph: the way the proposed model is selected is also my main criticism. All alternative models do not fit the data. Therefore, the proposed model is suggested. However, I do not grasp any direct support for this model. Either I missed it or it is not presented.

Concerning the specific model in Figure 5, the reviewer is correct. We do not provide direct evidence for a side-ways interaction. However, we have evidence of transient interactions and our data rule out several scenarios of interaction, leaving 5C as the most likely model. This is also the main conclusion of this paper: In conclusion, the SBDs of OpuA transiently interact in a docking competent conformation, explaining the cooperativity between the SBDs during transport. The conformation of this interaction is not fixed but differs substantially between different conditions.

Because the interaction is very short-lived it was not possible to visualize molecular details of this interaction. We present Figure 5 to hypothesize the most likely type of interaction, since many possibilities can be excluded with the vast amount of presented data. To make our point more clear that we discuss models and rule out several possibilities but not demonstrate a specific interaction between the SBDs, we now write on page 10 (changes marked in bold): We have shown that the SBDs of OpuA come close together in a short-lived state, which is responsive to the addition of glycine betaine (Figure 4A). Although the occurrence of the state varies between different conditions, it was not possible to negate the high-FRET state completely, not even under very high or low KCl concentrations, or in the presence of 50 mM arginine plus 50 mM glutamate (Figure 4A,B). To evaluate possible interdomain interactions scenarios we consider the following: (1) The SBDs of OpuA are connected to the TMDs with very short linkers of approximately 4 nm, which limit their movement and allow the receptor to sample a relatively small volume near its docking site. (2) in low ionic strength condition OpuA-K521C displays a high FRET state with mean FRET values of 0.7-0.8, which correspond to inter-dye distances of approximately 4 nm. (3) The high FRET state is responsive to glycine betaine, which points toward direct communication between the two SBDs. (4) The distance between the density centers of the SBDs in the cryo-EM reconstructions (based on particles with a low and high FRET state) is 6 nm, which aligns with the dimensions of an SBD (length: ~6 nm, maximal width: ~4 nm). These findings collectively indicate that two SBDs interact but not necessarily in a singular conformation but possibly as an ensemble of weakly interacting states. Hence, we discuss three possible SBD-SBD interaction models to explain the highFRET state:

**Reviewer #2 (Recommendations For The Authors):**
In the abstract and elsewhere the authors suggest that the SBDs physically interact with one another, and that this interaction is important for the transport mechanism, specifically for its cooperativity.I feel that this main claim is not well established. The authors convincingly demonstrate that the SBDs largely occupy two states relative to one another and that in one of these states, they are closer than in the other. Unless I have missed (or failed to understand) some major details of the results, I did not find any evidence of a physical interaction. Have the authors established that the high FRET state indeed corresponds to the physical engagement of the SBDs? I feel that a direct demonstration of an interaction is much missing.Along the same lines, in the low-salt cryo-EM structure, where the SBDs are relatively closer together, the SBDs are still separated and do not interact.

See also our response to the final comment of reviewer 1. Furthermore, please carefully consider the following: (1) FRET values of 0.7-0.8 correspond to inter-dye distances of approximately 4 nm. (2) The high FRET state is responsive to glycine betaine, which points toward direct communication between the two SBDs. (3) The cryo-EM reconstruction is the average of all the particles in the final dataset, including both the particles with a low and high FRET state. Further, the local resolution of the SBDs in the cryo-EM map is low, indicative of high degree of flexibility. Thus, a potential interaction is possible within the observed range of flexibility. (4) The distance between the density centers is 6 nm, aligning with the dimensions of an SBD (length: 6 nm, maximal width: 4 nm). These factors collectively indicate SBD interactions, and we present these points now more explicitly in Figure 4 and the last part of the results section (page 9).

Once the authors successfully demonstrate that direct physical interaction indeed occurs, they will need to provide data that places it in the context of the transport cycle. Do the SBDs swap ligand molecules between them? Do they bind the ligand and/or the transporter cooperatively? What is the role of this interaction?

We acknowledge the intriguing nature of the posed questions, but they extend beyond the scope of this study. It is extremely challenging to obtain high-resolution structures of highly dynamic multidomain proteins, like OpuA, and to probe transient interactions as we do here for the SBDs of OpuA. We therefore combined cryo-TEM with smFRET studies and perform the most advanced and state-of-theart analysis tools as acknowledged by reviewer 1. We link our observations on the structural dynamics and interactions of the SBDs to a previous study, where we showed that the two SBDs of OpuA interact cooperatively. We do not have further evidence that connect the physical interactions to the transport cycle. In our view, the collective datasets indicate that the here reported physical interactions between the SBDs increase the transport efficiency.

As far as I understand, the smFRET data have been interpreted on the basis of a negative observation, i.e., that it is "likely" that none of the FRET states corresponds to a docked SBD. To convincingly show this, a positive observation is required, i.e., observation of a docked state.

The aim of this study was to study interdomain dynamics and not specifically docking. We have previously shown that docking can be visualized via cryo-EM (Sikkema et al., 2020), however the SBDs of OpuA appear to only dock in specific turnover conditions. We now show that the high FRET state of OpuA cannot represent a docked state, but that the SBDs transiently interact (see our response to the first comment). Importantly, a docked state was also not found in the cryo-EM reconstructions at low ionic strength, representing the smFRET conditions where we observe the interactions between the SBDs. The high FRET state occupies 30% of the dwells in this condition, and such a high percentage of molecules would have become apparent during cryo-EM 3D classification in case they would form a docked state. Therefore, we conclude that docking does not occur in low ionic strength apo condition. We discuss this point and our reasoning on page 11 of the revised manuscript.

In this respect, I find it troubling that in none of the tested conditions, the authors observed a FRET state which corresponds to the docked state. Such a state, which must exist for transport to occur (as mentioned in the authors' previous publications), needs to be demonstrated. This brings me to my next question: why have the authors not measured FRET between the SBDs and the transporter? Isn't this a very important piece that is missing from their puzzle?

We agree that investigating docking behavior under varied turnover conditions requires focused experiments on FRET dynamics between the SBDs and the transporter. As noted on page 5, OpuA exists as a homodimer, implying that a single cysteine mutation introduces two cysteines in a single functional transporter. To specifically implement a cysteine mutation in only one SBD and one transmembrane domain, it is necessary to artificially construct a heterodimer. We recently published initial attempts in this direction, and this will be a subject for future research but still requires years of work.

Additionally, I feel that important controls are missing. For example, how will the data presented in Fig1 look if the transporter is labeled with acceptor or donor only? How do soluble SBDs behave?

In the employed labeling method, donor and acceptor dyes are mixed in a 1:1 ratio and randomly attached to the two cysteines in the transporter. This automatically yields significant fractions of donor only and acceptor only transporters which are always present during the smFRET recordings. We can visualize those molecules on the basis of the dye stoichiometry, which we calculate by using three types of photon counts: donor-based donor emission (FDD), donor-based acceptor emission (FDA) and acceptorbased acceptor emission (FAA).S=FDA+FDDFDA+FDD+FAA

Unfiltered plots look as follows (a dataset of OpuA-K521C at 600 mM KCl):

Donor only and acceptor only molecules have a very well discernible stoichiometry of 1 and 0, respectively. The filtering procedure is described in the materials and methods section, and these plots can be found in the supplementary database. We did not add them to the main text or supplementary materials of the original manuscript, as this is a very common procedure in the field of smFRET. We now include such a dataset in the revised manuscript.

Soluble SBDs of OpuA have been studied previously (e.g. Wolters et al., 2010 & De Boer et al. 2019). For example, we have shown by SEC-MALLS that soluble SBDs do not form dimers, which is consistent with our notion that the SBDs interact with low affinity. It is not possible to study interdomain dynamics between soluble SBDs by smFRET, because the measurements are carried out at picomolar concentrations (monomeric conditions). We emphasize that smFRET measurements with native complexes, with SBDs near each other at apparent millimolar concentrations, is physiologically more relevant.

Additional comments:(1) "It could well be that cooperativity and transient interactions between SBDs is more common than previously anticipated" and a similar statement in the abstract. What evidence is there to suggest that the transient interactions between SBDs are a common phenomenon?

On page 11, we write: Dimer formation of SBPs has been described for a variety of proteins from different structural clusters of substrate-binding proteins [33–38,51–53].We cite 9 papers that report SBD/SBP dimers. This suggest to us that the phenomenon of interacting substrate-binding proteins could be more common. Moreover, the concentration of maltose-binding protein and other SBPs in the periplasm of Gram-negative bacteria can reach (sub)millimolar concentrations, and low-affinity interactions may play a role not only in membrane protein-tethered SBDs (like in OpuA) but also be important in soluble substrate-receptors. Such low-affinity interactions are rarely studied in biochemical experiments.

(2) I think that the data presented in 1B-C better suits the supplementary information.

Figure 1B-D is already a summary of the supplementary information that describes the optimization of OpuA purification. We think it is valuable to show this part of the figure in the main text. A very clean and highly pure OpuA sample is essential for smFRET experiments. Quality of protein preparations and data analysis are key for the type of measurements we report in this paper.

(3) "the first peak in the SEC profile corresponds...." The peaks should be numbered in the figure to facilitate their identification.

We have changed the figure as suggested.

(4) "smFRET is a powerful tool for studying protein dynamics, but it has only been used for a handful of membrane proteins". With the growing list of membrane proteins studied by smFRET I find this an overstatement.

We removed this sentence in the new version of the manuscript.

(5) "We rationalized that docking of one SBD could induce a distance shift between the two SBDs in the FRET range of 3-10 nm (Figure 1E)" How and why was this assumed?

We realize that this is one of the sentences that caused confusion about the aim of this study. In this part of the manuscript, we should not have used docking as an example and we apologize for that. We replaced the sentence by: These variants are used to study inter-SBD dynamics in the FRET range of 310 nm (Figure 1E).

Also Figure 1E was adjusted to prevent confusion:

**Author response image 2. sa3fig2:** 

In addition, to avoid any confusion we changed the following sentence on page 4 (changes marked in bold): We designed cysteine mutations in the SBD of OpuA to study interdomain dynamics in the full length transporter.

(6) "However, the FRET distributions are broader than would be expected from a single FRET state, especially for OpuA-K521C" Have the authors established how a single state FRET of OpuA looks? Is there a control that supports this claim?

Below we compare two datasets from OpuA-K521C in 600 mM KCl with a typical smFRET dataset from the well-studied substrate-binding protein MBP from *E. coli*, which resides in a single state. Left: OpuA-K521C; Right: MBP

**Author response image 3. sa3fig3:** 

We agree that this cannot be assumed from the presented data. Therefore we rewrote this sentence: However, the FRET distributions tail towards higher FRET values, especially OpuA-K521C.

(7) "V149Q was designed as a mild mutation that would reduce docking efficiency and thereby substrate loading, but leave the intrinsic transport and ATP hydrolysis efficiency intact." I find this statement confusing: How can a mutation reduce docking efficiency yet leave the transport activity unchanged?

We rewrote the sentences (changes marked in bold): V149Q was designed as a mild mutation that would reduce docking efficiency and thereby substrate loading, but leave the ionic strength sensing in the NBD and the binding of glycine betaine and ATP intact. Accordingly, a reduced docking efficiency should result in a lower absolute glycine betaine-dependent ATPase activity. At the same time the responsiveness of the system to varying KCl, glycine betaine, or Mg-ATP concentrations should not change.

(8) Along the same lines: "whereas the glycine betaine-, Mg-ATP-, or KCl-dependent activity profiles remain unchanged" vs. "OpuA-V149Q-K521C exhibited a 2- to 3-fold reduction in glycine betainedependent ATPase activity".

See comment at point 7.

(9) In general, I find the writing wanting at places, not on par with the high standards set by previous publications of this group.

We recognize the potential ambiguity in our phrasing. We hope that after incorporating the feedback provided by the reviewers our manuscript will convey our findings in a clearer manner.

Extra changes to the text:

(1) Title changed: The substrate-binding domains of the osmoregulatory ABC importer OpuA physically transiently interact

(2) Second part of the abstract changed: We now show, by means of solution-based single-molecule FRET and analysis with multi-parameter photon-by-photon hidden Markov modeling, that the SBDs transiently interact in an ionic strength-dependent manner. The smFRET data are in accordance with the apparent cooperativity in transport and supported by new cryo-EM data of OpuA. We propose that the physical interactions between SBDs and cooperativity in substrate delivery are part of the transport mechanism.

(3) Page 6, third paragraph and Figure 2B: the wrong rate number was extracted from table 1. Changed this in the text and figure: 112 s-1 173 s-1. It did not affect any of the interpretations or conclusions.

(4) Page 8, last paragraph, changed: smFRET was also performed in the absence of KCl and with a saturating concentration of glycine betaine (100 µM). The mean FRET efficiency of the highFRET state of OpuA-K521C increased to 0.78, which corresponds to an inter-dye distance of about 4 nm. This indicates that the dyes at the two SBDs move very close towards each other (Figure 4A) (Table 1) (Supplementary File 34).

(5) Page 9, second paragraph changed: Due to the inherent flexibility of the SBDs, with respect to both the MSP protein of the nanodisc and the TMDs of OpuA, their resolution is limited. Furthermore, the cryo-EM reconstructions average all the particles in the final dataset, including those with a low and high FRET state. Nevertheless, in both conditions, the densities that correspond to the SBDs can be observed in close proximity (Figure 4D). The distance between the density centers is 6 nm and align with the dimensions of an SBD, providing further evidence for physical interactions between the SBDs.